# How daily groundwater table drawdown affects the diel rhythm of hyporheic exchange

Liwen Wu[1,2], Jesus D. Gomez-Velez[3,4], Stefan Krause[5,6], Anders Wörman[7], Tanu Singh[5,8], Gunnar Nützmann[1,2], and Jörg Lewandowski[1,2]

[1]Department of Ecohydrology, Leibniz-Institute of Freshwater Ecology and Inland Fisheries (IGB), Berlin, Germany

[2]Geography Department, Humboldt-University, Berlin, Germany

[3]Department of Civil and Environmental Engineering, Vanderbilt University, Nashville, TN, USA

[4]Department of Earth & Environmental Sciences, Vanderbilt University, Nashville, TN, USA

[5]School of Geography, Earth and Environmental Sciences, University of Birmingham, UK

[6]LEHNA-Laboratory of Ecology of Natural and Man-Impacted Hydrosystems, University Claude Bernard Lyon 1, Lyon, France

[7]Division of River Engineering, KTH-Royal Institute of Technology, Stockholm, Sweden

[8]Now at Department of Numerical Mathematics, Technical University of Munich, Garching, Germany

**Correspondence:** Liwen Wu (liwen.wu@igb-berlin.de)

**Abstract.** Groundwater table dynamics extensively modify the volume of the hyporheic zone and the rate of hyporheic exchange processes. Understanding the effects of daily groundwater table fluctuations on the tightly coupled flow and heat transport within hyporheic zones is crucial for water resources management. With this aim in mind, a physically based model is used to explore hyporheic responses to varying groundwater table fluctuation scenarios. Effects of different timing and amplitude of groundwater table daily drawdowns under gaining and losing conditions are explored in hyporheic zones influenced by natural flood events and diel river temperature fluctuations. We find that both diel river temperature fluctuations and daily groundwater table drawdowns play important roles in determining the spatiotemporal variability of hyporheic exchange rates, temperature of exfiltrating hyporheic fluxes, mean residence times, and hyporheic denitrification potentials. Groundwater table dynamics present substantially distinct impacts on hyporheic exchange under gaining or losing conditions. The timing of groundwater table drawdown has a direct influence on hyporheic exchange rates and hyporheic buffering capacity on thermal disturbances. Consequently, the selection of aquifer pumping regimes has significant impacts on the dispersal of pollutants in the aquifer and thermal heterogeneity in the sediment.

## 1    Introduction

Hyporheic zones are transitional areas between surface water and groundwater environments, which often exhibit marked physical, chemical, and biological gradients that drive the exchanges of water flow, energy, solute and microorganisms between surface and subsurface regions (Boano et al., 2014). Although the hyporheic zone is a small veneer, it has disproportionately
significant effects on nutrient cycling and river ecological functioning (Malcolm et al., 2002; Krause et al., 2009; Gomez-Velez et al., 2015). Understanding the spatiotemporal variability of hyporheic exchange processes is key to characterizing the nutrient cycling and river ecosystem functioning (Lewandowski et al., 2019).

Hydrological drivers and modulators of time-varying hyporheic exchange processes have been extensively studied in the last decade. The hydraulic gradient as the main driver of hyporheic exchange processes is changing along the sediment-
water interface, determining (1) the spatiotemporal variability of hyporheic zone extents and (2) characteristic time scales of hyporheic exchange (Boano et al., 2013; Ward et al., 2017; Gomez-Velez et al., 2017). Factors influencing the hydraulic gradient at the sediment-water interface include channel flow (Trauth and Fleckenstein, 2017; Grant et al., 2018; Broecker et al., 2018; Singh et al., 2020), geomorphological settings (Tonina and Buffington, 2011; Schmadel et al., 2016; Singh et al., 2019; Chow et al., 2019), and regional groundwater flow (Nützmann et al., 2014; Malzone et al., 2016; Wu et al., 2018). Sediment and
fluid properties do not drive hyporheic exchange, but they modulate hyporheic exchange substantially: sediment heterogeneity can alter hyporheic flow paths and residence time distributions, creating hot spots for biogeochemical transformations (Sawyer and Cardenas, 2009; Gomez-Velez et al., 2014; Pescimoro et al., 2019; Chow et al., 2020; Earon et al., 2020); fluid properties, i.e., density and viscosity, are functions of temperature and directly influence the hydraulic conductivity, thus hyporheic flow. Consequently, river temperature variability (i.e., diel and seasonal river temperature fluctuations) induces significant changes
of hyporheic exchange processes (Cardenas and Wilson, 2007a). The spatiotemporal variability of the drivers and modulators eventually results in dynamic hyporheic exchange processes. Among these drivers and modulators, the combined effects of regional groundwater flow and river temperature on dynamic hyporheic exchanges are comparably understudied.

Depending on the direction of net groundwater flow, the river can be gaining when groundwater discharges into the river, or losing when river recharges the aquifer (Winter et al., 1998) (Fig. 1a). Different directions of groundwater flow result in
substantially different flow fields (Fig. 1b and 1c). Large groundwater upwelling and downwelling may compress hyporheic zone's spatial extent and reduce the hyporheic exchange flow rate. Nevertheless, most of the previous numerical modeling studies about the impact of groundwater direction on hyporheic exchanges are either limited to steady hydrological conditions, and/or a uniform groundwater flow conditions (Cardenas and Wilson, 2006, 2007b; Boano et al., 2008; Trauth et al., 2013; Marzadri et al., 2016; Wu et al., 2018). Although there are recent field investigations on the role of transient groundwater table

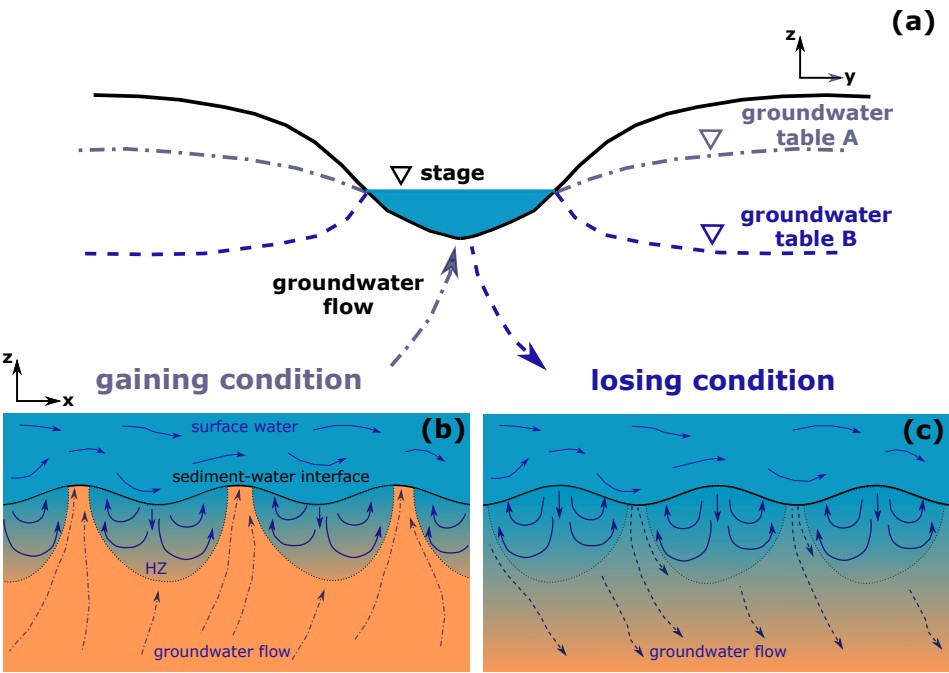

**Figure 1.** Schematic description of (a) gaining and losing groundwater systems and bedform-induced hyporheic exchanges under (b) gaining and (c) losing conditions. The river can be gaining when groundwater discharges into the river (scenario of groundwater table A), or losing when river recharges the aquifer (scenario of groundwater table B). Different directions of groundwater flow result in substantially different flow field, location and geometry of hyporheic zones.

fluctuations in hyporheic exchange processes (Malcolm et al., 2006; Ward et al., 2013; Zimmer and Lautz, 2014), they usually lack a quantification of the impact of groundwater table dynamics on hyporheic exchange processes (Malzone et al., 2016).

Groundwater table fluctuations are observed across multiple temporal scales. On seasonal scales, rainfall and irrigation pumping following well-defined seasonal cycles cause groundwater table fluctuations; on daily scales, phreatophytes (long-rooted plants that take up water from the saturated zone) induced water-use and anthropogenic pumping activities are the

main causes for groundwater table fluctuations; on event-scales, groundwater tables fluctuate in response to storm events (Todd and Mays, 2005; Butler Jr et al., 2007; Malzone et al., 2016). Both numerical modeling studies and field observation indicate that groundwater table fluctuations have a significant control on the hydraulic gradient change at the sediment-water interfaces, which is the main driver of transient hyporheic responses (Malcolm et al., 2006; Voltz et al., 2013; Malzone et al., 2016). However, these studies are usually focused on seasonal and event-scale groundwater table fluctuations. The role of daily

groundwater table fluctuations for hyporheic exchange processes requires more attention.

River temperature often fluctuates with a clear daily cycle in response to the diurnal change in solar radiation (Caissie, 2006). This daily change in river temperature directly affects water viscosity and density, and subsequently the hydraulic conductivity

of the sediment. As a consequence, hyporheic exchange rates often exhibit a diel fluctuation pattern due to the temperature-dependent hydraulic conductivity that governs the flow transport in the sediment. Wu et al. (2020) observed that hyporheic

exchange fluxes inherit the daily-scale spectral signatures from river temperature fluctuations, and noticeably, however, these signatures are absent in river discharge of the studied site. This observation evidently indicates a strong control of the diel river temperature fluctuation on hyporheic exchange processes. However, the temperature-dependent diel rhythm of hyporheic exchange rates can be interfered by the daily groundwater table fluctuations due to evapotranspiration and anthropogenic pumping activities. Therefore, understanding the two players, namely daily groundwater hydraulic gradient change (as a result

of daily groundwater table fluctuations) and diel hydraulic conductivity change (as a result of diel river temperature fluctuation), is important to characterize dynamic hyporheic exchange processes.

In the present study, we aim to quantify the impact of river temperature fluctuations and groundwater table drawdown on hyporheic exchange processes at daily scales, as well as to better understand implications on hyporheic zone's potential for denitrification and thermal buffering. With these objectives in mind, a series of synthetic groundwater scenarios corresponding

to different timings of groundwater table drawdown under gaining and losing conditions is applied in a physically based hyporheic flow and heat transport model. Hyporheic exchange rates, temperature distribution and denitrification efficiency are quantified to assess the impacts of river temperature and groundwater level fluctuations on hyporheic exchange processes. Our findings for the first time provide insights into the dynamic hyporheic responses to impacts of daily groundwater withdrawal and river temperature fluctuations, allowing for a better mechanistic understanding on hyporheic exchange processes and hence

an improved pumping operational scheme.

## 2   Methods

### 2.1   Model Domain

To understand the hyporheic exchange in response to changing river discharge, temperature and groundwater table fluctuations, a two-dimensional conceptualization is proposed based on Wu et al. (2018) and Wu et al. (2020) (Fig. 2a). The sediment

is assumed homogeneous and isotropic with a sinusoidal sediment-water interface of wavelength $\lambda$ and amplitude $\Delta$, representing periodic bedforms. The streamwise length ($L$) is $3\lambda$ and the depth of the model domain ($d_{gw}$) is $5\lambda$, respectively. Bedforms are assumed stationary and fully saturated. Transport of flow, solute, and heat is simulated by using COMSOL Multiphysics (version: 5.4) with finite element method using a mesh with telescopic refinement near the boundaries and approximately 54,000 elements. The simulations are mesh-independent. The computation time for a full-length scenario is around 60 hours.

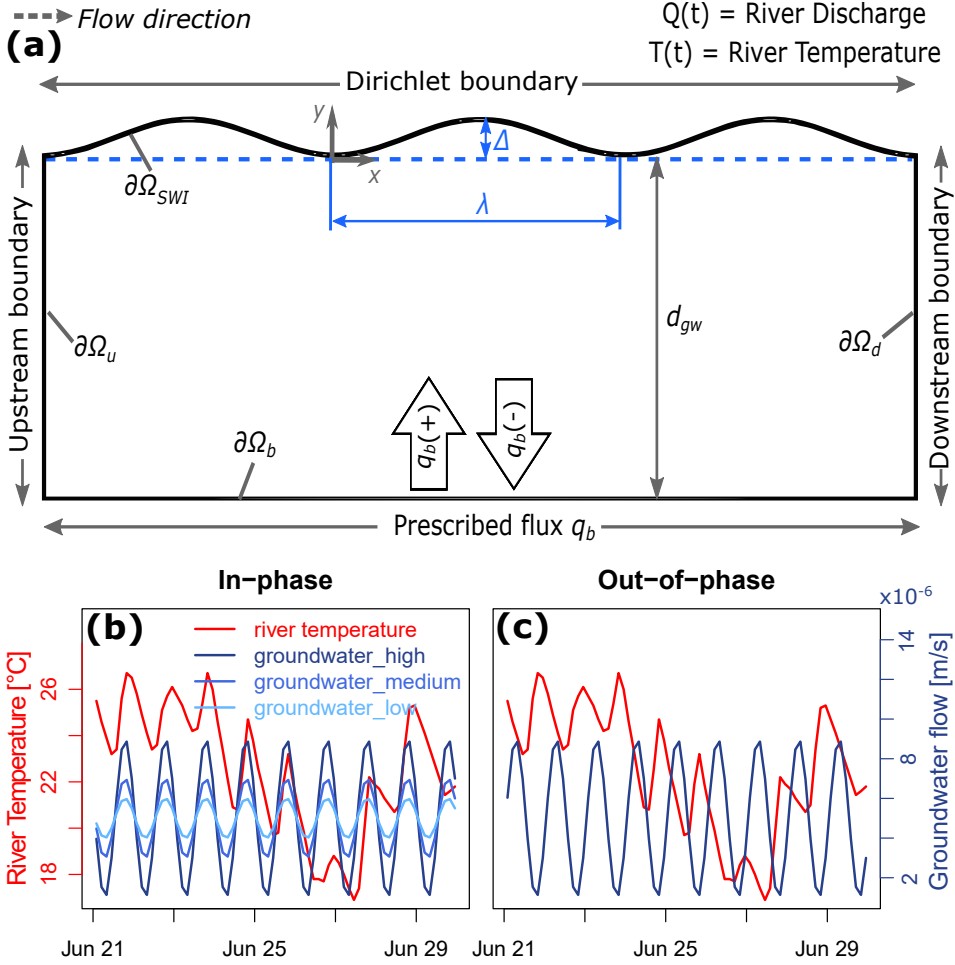

**Figure 2.** Model geometry and scenarios. (a) Schematic representation of the sediment domain. The top boundary is sinusoidal with amplitude $\Delta$ and wavelength $\lambda$. Lateral boundaries are periodic, representing an infinite domain in the longitudinal direction. Groundwater enters (gaining condition, $q_b(+)$) or leaves (losing condition, $q_b(-)$) the domain through the bottom boundary. (b) In-phase groundwater conditions with three amplitudes of groundwater level fluctuations. In-phase condition means that the strongest groundwater fluxes occur around the same time of the day as the highest river temperature. (c) Out-of-phase groundwater condition, i.e. strongest groundwater fluxes occur almost simultaneously to lowest river temperatures. Temperature time series are obtained from the U.S. Geological Survey (USGS, Site ID: 06893970). Groundwater flux is conceptualized as sinusoidal curves with varying amplitudes representing the strength of the groundwater upwelling or downwelling, and varying phases representing in-phase and out-out-phase scenarios. For figure clarity, a 10-day time window is selected arbitrarily from Jun 21 to Jun 30, 2017.

## 2.2 Model for Coupled Flow and Heat Transport

### 2.2.1 Model for Groundwater Flow

Groundwater flow is described using Darcy's law in a non-deformable porous media (Bear, 1972). The top boundary is a Dirichlet boundary. Lateral boundaries are periodic boundaries, representing an infinite domain in the longitudinal direction. The bottom boundary is either prescribed inflow for groundwater gaining condition ($q_b(+)$) or outflow for groundwater losing condition ($q_b(-)$).

$$\theta \frac{\partial \rho}{\partial t} = \nabla \cdot \left[ \rho \frac{\kappa}{\mu} (\nabla p + \rho g \nabla h) \right] \tag{1a}$$

$$p(x, y = Z_{bed}(x), t) = \rho g \, h_{SWI}(x, t) \text{ for } \partial \Omega_{SWI} \tag{1b}$$

$$p(x = -\lambda, y, t) = p(x = 2\lambda, y, t) + \rho g[h_{SWI}(x = -\lambda, t) + h_{SWI}(x = 2\lambda, t)] \text{ for } \partial \Omega_u \text{ and } \partial \Omega_d \tag{1c}$$

$$\mathbf{n} \cdot \left[ -\frac{\kappa}{\mu} (\nabla p + \rho g \nabla z) \right] = -q_b \text{ for } \partial \Omega_b \tag{1d}$$

where $t$ is time [T], $\theta$ is porosity [-] as 0.3, $p(\mathbf{x}, t)$ is pressure [ML$^{-1}$T$^{-2}$], $g$ is gravitational acceleration [LT$^{-2}$], $\kappa$ is permeability [L$^2$] as $1E-10$ m$^2$, $\rho$ is fluid density [ML$^{-3}$], $\mu$ is fluid dynamic viscosity [ML$^{-1}$T$^{-1}$], Darcy velocity is $\mathbf{q} = -\frac{\kappa}{\mu}(\nabla p + \rho g \nabla h))$ [LT$^{-1}$], $Z_{bed}(x) = (\Delta/2)\sin(2\pi x/\lambda)$ is the elevation of the water-sediment interface [L], $\mathbf{n}$ is an outward vector normal to the boundary [-], $q_b$ is groundwater flux [LT$^{-1}$].

Prescribed head distributions are applied at the sediment-water interface (Wörman et al., 2006)

$$h_{SWI}(x, t) = H_s(t) - Z_{bed}(x) + \frac{2 h_d(t)}{\Delta} Z_{bed}\left(x + \frac{\lambda}{4}\right) \tag{2}$$

where $H_s(t)$ [L] is the transient river stage, and $h_d(t)$ is the dynamic head fluctuations (Fehlman, 1985; Elliott and Brooks, 1997)

$$h_d(t) = 0.28 \frac{U_s(t)^2}{2g} \begin{cases} \left( \dfrac{\Delta}{0.34 \, H_s(t)} \right)^{3/8} & \text{for } \dfrac{\Delta}{H_s(t)} \leqslant 0.34 \\[3mm] \left( \dfrac{\Delta}{0.34 \, H_s(t)} \right)^{3/2} & \text{for } \dfrac{\Delta}{H_s(t)} > 0.34 \end{cases} \tag{3}$$

with the mean velocity $U_s(t) = M^{-1} H_s(t)^{2/3} S^{1/2}$ estimated with the Chezy equation for a rectangular channel with slope $S$ [-] and Manning coefficient $M$ [L$^{-1/3}$T] (Dingman, 2009).

In the present study, an aspect ratio (the ratio between amplitude and wavelength $\Delta/\lambda$) of 0.1 and slope of 0.01 are used to describe the geomorphological setting as dunes (Dingman, 2009; Bridge, 2009). A Manning coefficient of 0.05 is chosen.

Although this two-dimensional conceptualization is simple in nature, it allows us to capture the hydrodynamic effects on hyporheic exchange based on empirical approaches. A comprehensive discussion on the effect of local morphology (i.e., aspect ratios), channel slope, and sediment heterogeneity on the transient hydraulic pressure propagation within hyporheic zones can be found in Wu et al. (2018).

### 2.2.2 Model for Heat Transport

Transport of heat in porous media is described by using the heat transport equation (Bejan, 1993; Nield and Bejan, 2013)

$$\frac{\partial T}{\partial t} = \nabla \cdot (\mathbf{D_T} \, \nabla T) - \nabla \cdot (\mathbf{v_T} \, T) \tag{4a}$$

$$T(x,t) = T_s \text{ for } \partial\Omega_{in,SWI} \tag{4b}$$

$$\mathbf{n} \cdot (\mathbf{D_T} \nabla T) = 0 \text{ for } \partial\Omega_{out,SWI} \tag{4c}$$

$$T(x = -L, y) = T(x = 2L, y) \text{ for } \partial\Omega_u \text{ and } \partial\Omega_d \tag{4d}$$

$$T(x,t) = T_b \text{ for } \partial\Omega_b \text{ under gaining condition} \tag{4e}$$

$$\mathbf{n} \cdot (\mathbf{D_T} \nabla T) = 0 \text{ for } \partial\Omega_b \text{ under losing condition} \tag{4f}$$

where $T$ is temperature [$\Theta$], $\mathbf{v_T} = (\rho_f \, c_f)/(\rho c)\mathbf{q}$ is the thermal front velocity [$LT^{-1}$], $\mathbf{D_T}$ is the hydrodynamic thermal dispersion tensor [$L^2T^{-1}$] calculated following Wu et al. (2020), and $\rho c = \theta \, \rho_f \, c_f + (1-\theta) \, \rho_s \, c_s$, is the specific volumetric heat capacity of the fluid-grains media [$ML^{-1}T^{-2}\Theta^{-1}$], $\rho_f \, c_f$ is the specific volumetric heat capacity of the fluid [$ML^{-1}T^{-2}\Theta^{-1}$], and $\rho_s \, c_s$ is the specific volumetric heat capacity of the solids [$ML^{-1}T^{-2}\Theta^{-1}$], $T_s$ is the temperature of the water column [$\Theta$], which is the measured river temperature time series. $\partial\Omega_{in,SWI}$ and $\partial\Omega_{out,SWI}$ represent the boundaries where surface water flows into and out of the sediment at the sediment-water interface, respectively. A mixed Dirichlet and Neumann boundary is used for heat transport along the sediment-water interface. Temperature at the bottom boundary is prescribed under gaining conditions. In this case, seasonal variations in groundwater temperature ($T_b$) are assumed sinusoidal with the mean of 10 °C and the amplitude of 3 °C. $T_b$ is higher than $T_s$ in winter and lower than $T_s$ in summer. Under losing conditions, the bottom boundary is represented by a pure convection of heat boundary.

### 2.2.3 Coupling Groundwater Flow and Heat Transport

Transport of flow and heat in porous media is coupled by the equations of state for density and viscosity (Furbish, 1996)

$$\mu(T) = m_5 T^5 + m_4 T^4 + m_3 T^3 + m_2 T^2 + m_1 T + m_0 \tag{5a}$$

$$\rho(T) = \rho_0 - \rho_0 \alpha (T - T_0) \tag{5b}$$

where viscosity is in Pa·s, temperature is in °C and $m_5 = -3.916 \times 10^{-13}$, $m_4 = 1.300 \times 10^{-10}$, $m_3 = -1.756 \times 10^{-8}$, $m_2 = 1.286 \times 10^{-6}$, $m_1 = -5.895 \times 10^{-5}$, and $m_0 = 1.786 \times 10^{-3}$. The reference density and temperature are $\rho_0 = 1000\,\text{kg/m}^3$ and $T_0 = 20\,°\text{C}$, respectively, and the thermal expansion coefficient is $\alpha = 2.067 \times 10^{-4}\,°\text{C}^{-1}$.

## 2.3 Model for Mean Residence Time

We use the mean residence time to describe the time that water is exposed to biogeochemical reactive sediments (Gomez-Velez and Wilson, 2013)

$$\theta \frac{\partial a_1}{\partial t} = \nabla \cdot (\mathbf{D} \nabla a_1) - \nabla \cdot (\mathbf{q} a_1) + \theta a_0 \tag{6a}$$

$$a_1(\mathbf{x}, t) = 0 \quad \text{for } \partial\Omega_{in,SWI} \tag{6b}$$

$$\mathbf{n} \cdot (\mathbf{D} \nabla a_1) = 0 \quad \text{for } \partial\Omega_{out,SWI} \tag{6c}$$

$$a_1(x_u, y, t) = a_1(x_d, y, t) \text{ for } \partial\Omega_u \text{ and } \partial\Omega_d \tag{6d}$$

$$a_1(\mathbf{x}, t) = a_{1b} \quad \text{on } \partial\Omega_b \text{ under gaining condition} \tag{6e}$$

$$\mathbf{n} \cdot (\mathbf{D} \nabla a_1) = 0 \quad \text{on } \partial\Omega_b \text{ under losing condition} \tag{6f}$$

where $a_1(\mathbf{x}, t)$ is the mean of the residence time distribution [T], $t$ is time [T], $\mathbf{x} = (x, y)$ is the spatial location vector, $\mathbf{q}$ is the Darcy flux [LT$^{-1}$], and $\mathbf{D}$ is the dispersion-diffusion tensor defined by (Bear, 1972), $a_0 = 1$ is the initial condition for the moments, $a_{1b}$ is the mean residence time of the groundwater fluid [T$^{-1}$]. $a_{1b}$ is prescribed, similar to Gomez-Velez et al. (2014), and a value of 10 years is assumed based on Mcguire and Mcdonnell (2006).

## 2.4 Defining Hyporheic Zones

In the present study, the hyporheic zone is defined as the sediment area containing at least 90% of the surface water (Triska et al., 1989; Gooseff, 2010). Numerical tracer is simulated with advection-dispersion equation simultaneously as the flow transport model to define the boundary of hyporheic zones

$$\theta \frac{\partial C}{\partial t} = \nabla \cdot (\mathbf{D} \nabla C) - \nabla \cdot (\mathbf{q} C) \tag{7}$$

where $C$ is the concentration of the non-reactive tracer[ML$^{-3}$], $\mathbf{q}$ is the Darcy flux [LT$^{-1}$], and $\mathbf{D} = \{D_{ij}\}$ is the dispersion-diffusion tensor defined as Bear (1972). The concentration of tracer in the surface water column is assumed as $C_s$. Therefore, the hyporheic zone is defined when $C \geq 0.9 C_s$ in the sediment. The boundary of the hyporheic zone is renewed at every time

point and therefore it's changing over time under varying flow conditions.With this condition, the threshold $C \geq 0.9C_s$ will be eventually exceeded across the entire domain under losing conditions. Therefore, hyporheic zone is tracked using reversed Darcy flow in order to identify the areas with the largest influence from the surface water under losing conditions. Using this definition, water flow into the hyporheic zone is defined as infiltrating hyporheic fluxes and water flow out of the hyporheic zone is defined as the exfiltrating hyporheic fluxes.

## 2.5 Study Scenarios

To better focus on the effect of river temperature and groundwater table dynamics on hyporheic exchange, we use the observed river discharge and temperature measurements from USGS gauging station (ID: 06893970). The gauging station is located in Spring Branch Creek at Holke Road in Independence, Missouri (Lat $39°05'18''$, Long $94°20'36''$ referenced to North American Datum of 1927). The station is on upstream left bank Missouri Highway 78 about 2.4 km above the confluence with the Little Blue River with a drainage area of 22 km$^2$. The observation period is from 2014-10-16 to 2017-10-16. Spectral analysis, presented in a previous study, shows that river temperature of this site has a clear daily fluctuation pattern; whereas the river discharge exhibits no daily fluctuations (the "reference site" in Fig. 5 presented in Wu et al. (2020)). Therefore, this site is an ideal site to explore the interactions of groundwater table dynamics and river temperature fluctuations on daily scales without the additional influence of daily river stage changes.

Daily groundwater table drawdown due to phreatophytes induced water-uptake mainly takes place in the afternoon when transpiration processes are strongest due to high air and river temperature; while agricultural, residential or industrial water-supply may cause water table drawdown at any time during the day. Since the objective of the present study is to explore the impacts of daily groundwater table drawdowns and diel river temperature fluctuations, the study focuses on two special cases: *in-phase* and *out-of-phase* conditions. In the in-phase condition, the highest hydraulic gradient between surface water and groundwater table (also means strongest groundwater flux) occurs around the same time of the day as the occurrence of the highest river temperature; in the out-of-phase condition, the highest hydraulic gradient between surface water and groundwater (also means strongest groundwater flux) occurs around the same time of the day as the occurrence of the lowest river temperature (Fig. 2b and 2c). Under gaining scenarios, out-of-phase conditions represent the natural state that highest air/river temperature occurs at the lowest water table (resulting to lowest groundwater flow rate) in the aquifer due to transpiration by vegetation; under losing scenarios, in-phase condition represents a scenario driven by transpiration, because the highest air/river temperature contributes to the strongest transpiration which results in a larger hydraulic head difference between river and aquifer, and thus contributes to the higher losing groundwater fluxes. The objective of this study is not to understand groundwater responses to pumping activities. Even though the timing of groundwater table drawdown depends on multiple factors, i.e. hydrological connectivity between wells and aquifer, aquifer properties for plant water-use, and pumping capacity and electricity tariff for anthropocentric pumping activities, the two special cases, namely in-phase and out-of-phase groundwater conditions, can capture the representative dynamic hyporheic responses to different timing of daily groundwater withdrawal under corresponding river temperature conditions.

Groundwater flow fluctuations, as a response to daily groundwater table drawdown, are conceptualized as sinusoidal curves with varying amplitudes and phases. Different phases reflect different timing of daily groundwater withdrawal, represented by the in-phase and out-of-phase groundwater flow conditions as described above. Different amplitudes represent different intensities of groundwater table drawdowns. For gaining system, three degrees of groundwater table fluctuation amplitudes are investigated. The highest fluctuation amplitude is two times higher than the scenario with medium amplitude, and four times higher than the scenario with low amplitude. Using the method proposed in Boano et al. (2008) which is described with details in the Supplementary Information, a change in the head difference ($dh$) of 3.5 cm is observed with the highest groundwater level fluctuation amplitude where $q_b$ varies daily from $1 \times 10^{-3}$ m/s to $9 \times 10^{-3}$ m/s. With the medium groundwater level fluctuation amplitude, the change in the head difference $dh$ is 1.8 cm. With the lowest groundwater level fluctuation amplitude, the change in the head difference $dh$ is 0.9 cm. These values are within a reasonable range for groundwater table fluctuations induced by plant water-use (Butler Jr et al., 2007). For simplicity, the same values of groundwater fluxes are also applied to losing systems.

No matter for plant's water-uptakes or anthropogenic activities (i.e., irrigation, municipal, or industrial water-supply), seasonal variations of groundwater fluxes cannot be neglected. For instance, a gradual transition of phreatophyte's dormancy in fall often induces a progressive diminishing in diurnal fluctuations and changes in the multi-day trend in groundwater tables (Butler Jr et al., 2007). Irrigation activities also follow the different seasonal water demand of agricultural plants. However, these seasonal changes are hard to generalize because groundwater flux variability depends on a variety of factors such as plant types, water availability and local climate conditions. Understanding the effect of seasonal groundwater variability is beyond the scope of the present study. Therefore, a uniform fluctuation amplitude of groundwater fluxes in the studied period is used.

# 3 Results

In the observation period, the river discharge is intermittent and characterized by short recession periods (approximately from 2 to 1500 m³/s); the river temperature shows clear seasonal variations (approximately from 0°C to 35°C) and daily fluctuations. Mean annual precipitation at the gauge location is 106 cm. Average annual air temperature at the gauge location is 12.6°C. There is no dam in the watershed.

## 3.1 Hyporheic Fluxes

### 3.1.1 Under Neutral Conditions

Under neutral condition, exfiltrating hyporheic fluxes (the red solid line in Fig. 3a) present similar temporal variations as infiltrating hyporheic fluxes (the black dotdash line in Fig. 3a). The diel fluctuations of exfiltrating hyporheic fluxes (the orange solid line in Fig. 3e and 3f) follow the diel river temperature fluctuations (the red solid line in Fig. 3e and 3f). In winter, when the river temperature (the red solid line in Fig. 3e) is relatively stable (around Jan 20), the exfiltrating hyporheic fluxes also

have negligible daily fluctuations; when temperature gets higher, the exfiltrating hyporheic fluxes start to fluctuate following the diel fluctuations of river temperature.

### 3.1.2 Under Gaining Conditions

Compared to neutral condition, groundwater upwelling leads to an increase of daily fluctuations of exfiltrating hyporheic fluxes. Under gaining condition, exfiltrating hyporheic fluxes (the red solid line in Fig. 3c) present larger daily amplitude variations than infiltrating hyporheic fluxes (the black dotdash line in Fig. 3c). These observations are reflected in the frequency domain using power spectrum. For neutral conditions, infiltrating and exfiltrating hyporheic fluxes show similar spectral power on both annual and daily scales (Fig. 3b); whereas for gaining conditions, the spectral power of exfiltrating hyporheic fluxes (the red solid line in Fig. 3d) at daily scales are markedly higher than the spectral power of infiltrating hyporheic fluxes (the black dotdash line in Fig. 3d).

With gaining groundwater fluxes, the fluctuation pattern of hyporheic fluxes changes substantially. Even with negligible diel fluctuations of river temperature (around Jan 20), the exfiltrating hyporheic fluxes still present clear daily fluctuations following the groundwater drawdown as indicated by the opposite fluctuating patterns between the exfiltrating hyporheic fluxes under in-phase (the black line in Fig. 3e and 3f) and out-of-phase (the blue line in Fig. 3e and 3f) groundwater scenarios. When temperature gets higher, the groundwater table-drawdown induced hyporheic fluctuations are maintained. The exfiltrating hyporheic fluxes under the in-phase scenario have an opposite fluctuation pattern with the exfiltrating hyporheic fluxes under the out-of-phase scenario, river temperature and the exfiltrating hyporheic fluxes under neutral condition; the exfiltrating hyporheic fluxes under the out-of-phase scenario fluctuate following river temperature. It's worth noticing that the peaks of exfiltrating hyporheic fluxes under out-of-phase scenario are slightly higher than the peaks of exfiltrating hyporheic fluxes under in-phase scenario at a warm temperature (Fig. 3f).

On Jul 27, under the same flood event, which causes a discharge increase from 2 to 1500 $m^3/s$ (the gray solid line in Fig. 3f), exfiltrating hyporheic fluxes increase much more under in-phase scenario (the black solid line) than under out-of-phase scenario (the blue solid line). The increase of exfiltrating hyporheic fluxes under in-phase scenario is nearly two times as high as the increase of hyporheic fluxes under out-of-phase scenario.

To explore the impact of groundwater table fluctuation amplitudes on dynamic hyporheic responses, groundwater table fluctuations with three different amplitudes are applied to simulate hyporheic exchange processes under in-phase scenarios (as the groundwater scenarios plotted in Fig. 2b). With the reduced groundwater upwelling amplitudes, the amplitudes of exfiltrating hyporheic flux fluctuations are also reduced (Fig. 4a). More than the amplitude reduction of exfiltrating hyporheic fluxes, with decreasing groundwater upwelling amplitude, the peaks of exfiltrating hyporheic fluxes (the black dash line, blue solid line and red solid line in Fig. 4b) are shifted towards the patterns which are more coinciding with diel river temperature fluctuations (the dash line in Fig. 4b) and hyporheic fluxes under neutral conditions (gray solid line). In other words, with

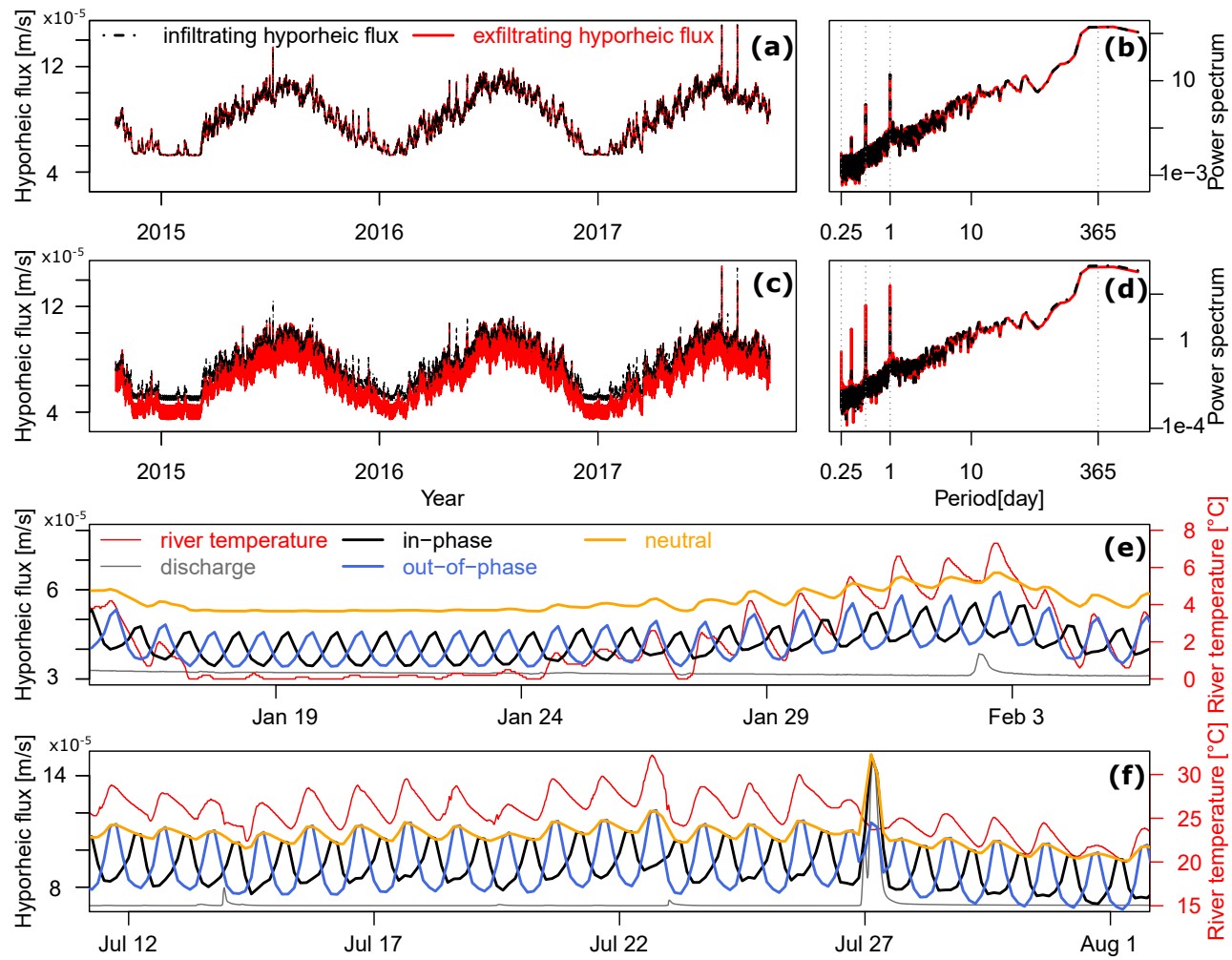

**Figure 3.** Effect of diel river temperature fluctuations and daily groundwater table drawdowns on hyporheic fluxes under gaining condition. Infiltrating and exfiltrating hyporheic fluxes under (a) neutral and (c) gaining conditions. Power spectrum of infiltrating and exfiltrating hyporheic fluxes under (b) neutral and (d) gaining conditions. Exfiltrating hyporheic fluxes under neutral conditions and under gaining conditions with in-phase and out-of-phase groundwater drawdown scenarios in (e) winter and (f) summer. For figure clarity, discharge is not scaled in e and f, but used only for qualitative comparisons. The flood event on Jul 27 causes a discharge increase from 2 to 1500 $m^3/s$.

decreasing groundwater table fluctuation amplitude, river temperatures exhibit stronger controls on the phase of hyporheic flux diel fluctuations.

Effects of groundwater table fluctuation amplitudes on dynamic hyporheic responses are only explored under in-phase scenarios, because under out-of-phase scenarios, fluctuations of exfiltrating hyporheic fluxes are almost always in the same phase with the diel river temperature fluctuations. Therefore, unlike in-phase scenarios, the phase shifts due to reduced amplitudes in groundwater table fluctuation are not observed. Reduced amplitudes in groundwater table fluctuation under out-of-phase scenarios only contribute to reduced amplitudes in exfiltrating hyporheic flux fluctuations. For simplicity, only results of in-phase scenarios are presented in Fig. 4.

### 3.1.3 Under Losing Conditions

Differing from the gaining conditions, under losing conditions, the fluctuation amplitudes of exfiltrating hyporheic fluxes are reduced compared with infiltrating hyporheic fluxes (Fig. 5a). This is also revealed in the frequency domain where the spectral power of exfiltrating hyporheic fluxes is reduced at the daily scales compared with the spectral power of infiltrating hyporheic fluxes (Fig. 5b).

The river temperature also demonstrates different impacts under losing conditions. In winter, when the river temperature (the red solid line in Fig. 5c) is relatively stable (around Jan 20), the exfiltrating hyporheic fluxes under in-phase and out-of-phase groundwater drawdown conditions exhibit an opposite fluctuation pattern resulting from the different timing of groundwater table drawdown (black and blue solid lines). This observation is the same with gaining conditions (Fig. 3e). However, when the river temperature gradually increases, the phase differences between the diel fluctuations of exfiltrating hyporheic fluxes under in-phase and out-of-phase scenarios are diminishing. In summer, when river temperature is relatively high, exfiltrating hyporheic fluxes under in-phase and out-of-phase conditions are fluctuating with almost the same phase with the river temperature (Fig. 5d). This observation is in great contrast to the gaining condition where the opposite fluctuation patterns between exfiltrating hyporheic fluxes under in-phase and out-of-phase conditions are kept from winter to summer (Fig. 3f).

Unlike gaining conditions, on Jul 27 under the same flood event (the gray solid line in Fig. 5d), the increases of exfiltrating hyporheic fluxes under in-phase and out-of-phase scenarios are similar. These distinctions indicate a vastly different coupled flow and heat transport pattern between gaining and losing systems.

## 3.2 Heat Transport in Hyporheic Zones

Snapshots of temperature distributions in the sediment demonstrate noticeable differences of the heat transport under different groundwater conditions in a summer day (2017-07-22 17:00) (Fig. 6). Under gaining conditions, both river and groundwater temperature play important roles in determining the temperature of the sediment; whereas under losing conditions, only the river temperature affects the temperature distributions in the sediment.

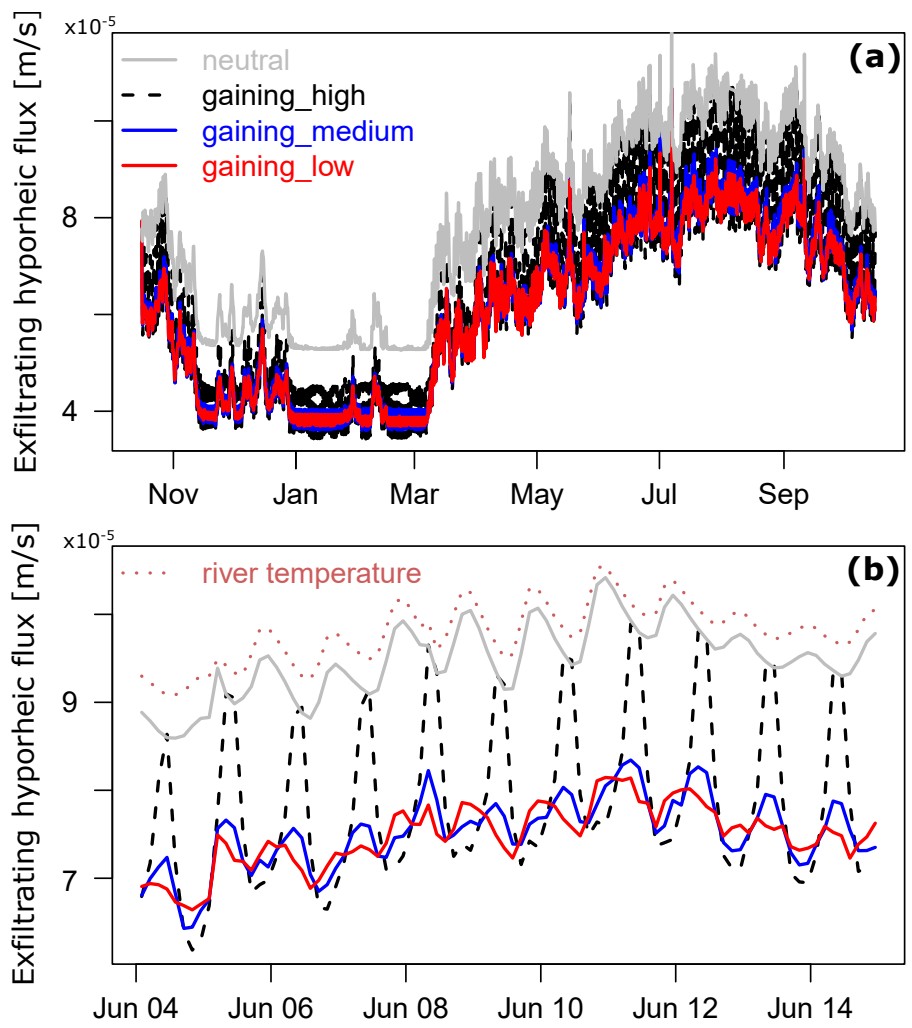

**Figure 4.** Effect of amplitudes in groundwater level fluctuations on hyporheic fluxes. (a) Exfiltrating hyporheic fluxes under neutral and gaining groundwater fluxes with three different amplitudes. (b) Comparisons of daily fluctuation phases among river temperature and exfiltrating hyporheic fluxes under neutral and gaining groundwater fluxes with three different amplitudes.

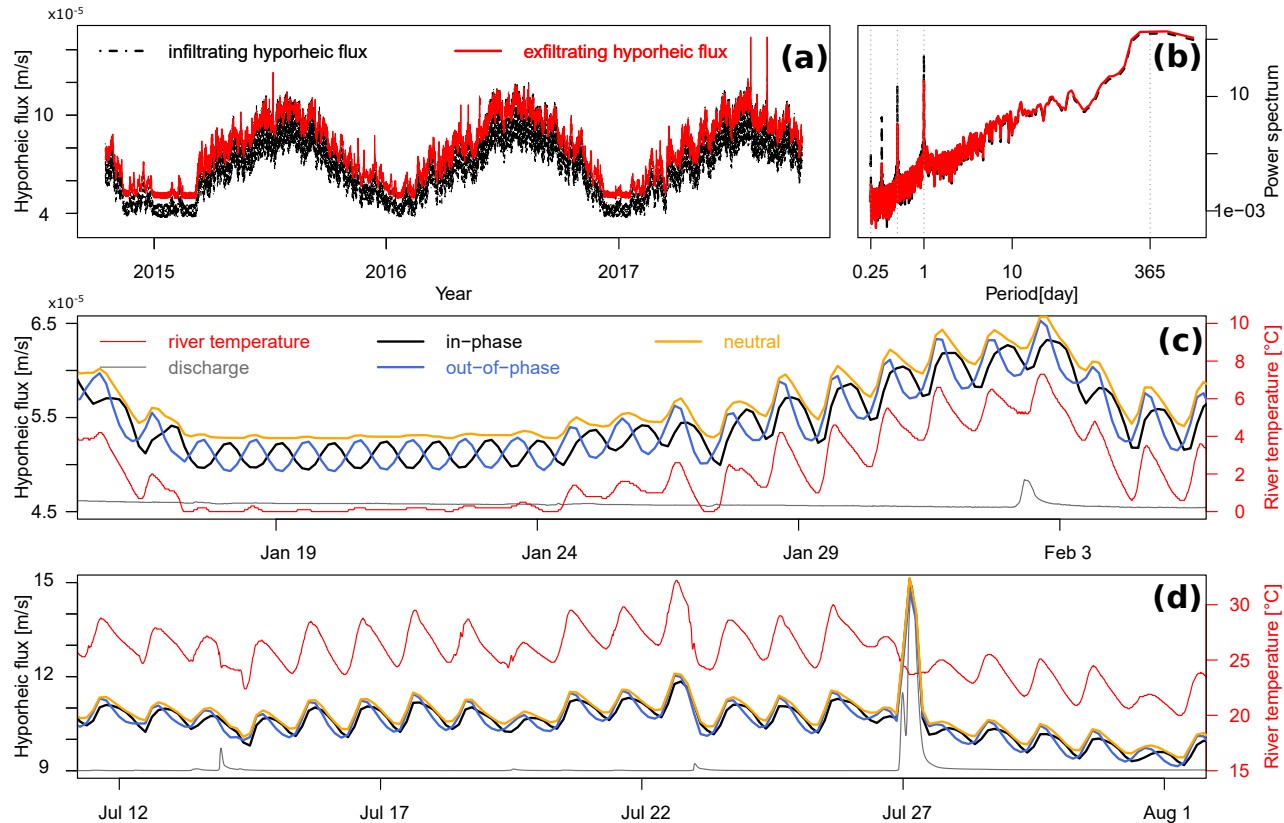

**Figure 5.** Effect of diel river temperature fluctuations and daily groundwater table drawdowns on hyporheic fluxes under losing condition. (a) Infiltrating and exfiltrating hyporheic fluxes under losing conditions and (b) corresponding power spectrum. Exfiltrating hyporheic fluxes under neutral conditions and under losing conditions with in-phase and out-of-phase groundwater drawdown scenarios in (c) winter and (d) summer. For figure clarity, discharge is not labeled in c and d. The flood event on Jul 27 causes a discharge increase from 2 to 1500 m$^3$/s

Temperature differences between river and exfiltrating hyporheic fluxes are explored for both gaining and losing, in-phase and out-of-phase conditions (Fig.7). Positive values indicate a higher river temperature than the temperature of exfiltrating hyporheic fluxes; negative values indicate a higher temperature of exfiltrating hyporheic fluxes. Under gaining conditions,
seasonal variations are observed for both in-phase and out-of-phase conditions. In winter, the exfiltrating hyporheic fluxes are generally warmer than the river; in summer, the river is generally warmer than the exfiltrating hyporheic fluxes. These seasonal variations are more prominent under out-of-phase conditions (the gray solid line in Fig. 7a) than under in-phase conditions (the blue dashed line in Fig. 7a). In summer, the exfiltrating hyporheic fluxes under out-of-phase conditions are much cooler than river water compared to the in-phase conditions. Under losing conditions, the differences between in-phase and out-of-phase
conditions are not as significant as under gaining conditions (Fig. 7b).

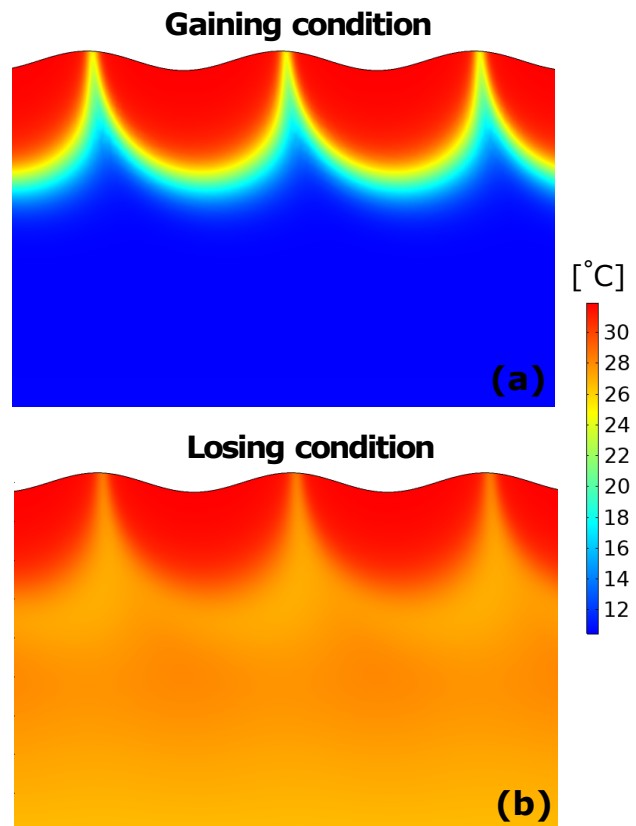

**Figure 6.** Snapshots of temperature distributions in the sediment on 2017-07-22 17:00 for (a) gaining and (b) losing groundwater conditions.

### 3.3   Reaction Significance Factor

Denitrification potential in hyporheic zones can be quantified using the reaction significance factor (RSF). The RSF is calculated as the ratio between hyporheic mean residence time and a characteristic time scale for denitrification, and then scaled by the proportion of the river discharge passing the hyporheic zone (Harvey et al., 2013). In the present study, we use the RSF calculated as the value per unit bedform area (denoted by the subscript "a")

$$\mathrm{RSF}_a = \frac{q_{HZ}}{Q} \cdot \frac{\tau_{HZ}}{\tau_{dn}} \tag{8}$$

where $q_{HZ}$ is the exfiltrating hyporheic fluxes [LT$^{-1}$], $Q$ is the river discharge [L$^3$T$^{-1}$], $\tau_{HZ}$ is the mean of the probability distribution of the residence time at any time point [T], $\tau_{dn}$ is the characteristic time scale for denitrification [T]. Typical time scales of denitrification in hyporheic zones are reported by Gomez-Velez and Harvey (2014); Gomez-Velez et al. (2015) and the quantiles are used in the calculation. The $25^{th}$, $50^{th}$, and $75^{th}$ quantiles are presented in Fig. S1 in the Supplementary

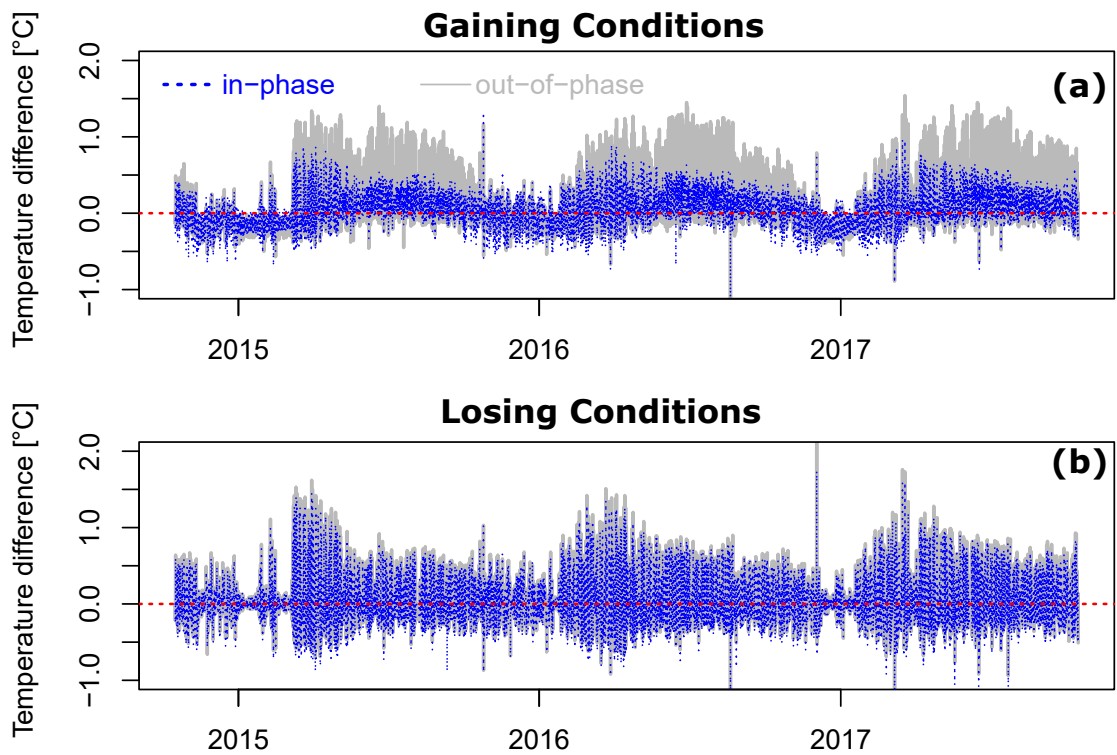

**Figure 7.** Temperature differences between river and exfiltrating hyporheic fluxes under (a) gaining and (b) losing in-phase and out-of-phase fluctuations of diel river temperature and daily groundwater table drawdown.

Information. It is worth noticing that instead of the denitrification, reaction potential of a different geochemical process can be assessed if a different characteristic time scale is applied in equation 8.

Under gaining conditions, $RSF_a$ displays opposite diel variations between in-phase and out-of-phase scenarios. Significant drops occur during flood events. Under losing conditions, $RSF_a$ is around 3.5 orders of magnitude lower than under gaining conditions. Daily-scale variations between in-phase and out-of-phase scenarios under losing conditions are less significant than under gaining conditions.

## 4 Discussion

### 4.1 Groundwater Modifies the Variability of Hyporheic Exchange Rates

With daily groundwater table drawdowns, additional hydraulic gradient changes on a daily scale contribute to enhanced diel fluctuations of hyporheic fluxes. Under neutral conditions, similar diel fluctuation patterns in both infiltrating and exfiltrating hyporheic fluxes (Fig. 3a and 3b) are mainly due to the change of hydraulic conductivity which is a function of diel temperature

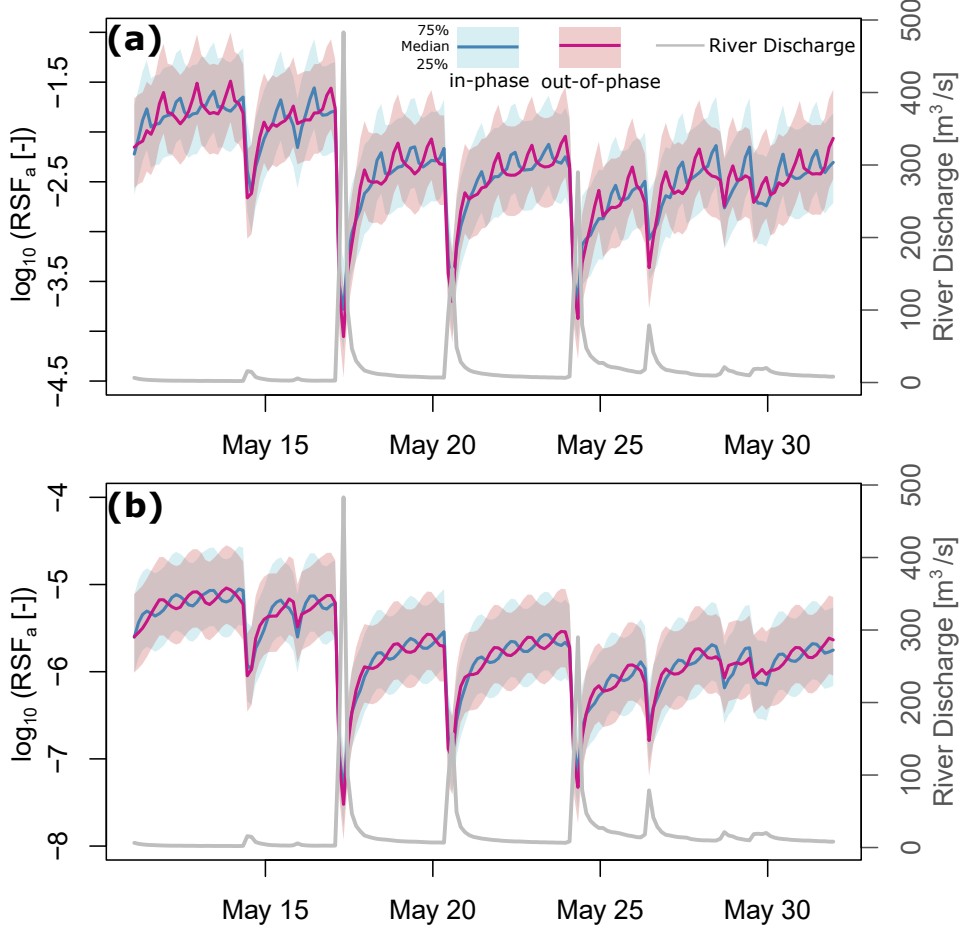

**Figure 8.** Reaction significance factors per unit area ($RSF_a$) for denitrification potentials from May 15 to May 30, 2017. (a) $RSF_a$ under gaining condition. (b) $RSF_a$ under losing condition. The results are selected arbitrarily with the considerations of figure clarity

.

fluctuations. Differing from neutral conditions, under gaining conditions the exfiltrating hyporheic fluxes show enhanced fluctuation amplitudes compared with the infiltrating hyporheic fluxes due to the additional fluctuations in the gaining groundwater fluxes that are mixed with the hyporheic fluxes which originate from the surface (Fig. 3c and 3d). Under losing conditions, the infiltrating hyporheic fluxes have higher fluctuation amplitudes because there is no mixing in the exfiltrating hyporheic fluxes under losing condition as the mixing occurred under gaining conditions according to the geochemical definitions of hyporheic zones (Fig. 5a and 5b). Under both gaining and losing conditions, the exfiltrating hyporheic fluxes exhibit higher fluctuation amplitudes than under neutral conditions, indicating groundwater table dynamics contribute to additional fluctuations in the hyporheic exchange fluxes.

The timing of groundwater table drawdown also affects hyporheic exchange rates. For instance, under the same flood event on July 27 (the gray solid line in Fig. 3f), the exfiltrating hyporheic flux under in-phase gaining conditions (the black solid line) increases more than the exfiltrating hyporheic flux under out-of-phase conditions (the blue solid line). This is because the groundwater gaining flux under the in-phase scenario is lowest in the course of the day when the flood arrives; whereas it is highest under the out-of-phase scenario. As a result of higher groundwater upward pressure, higher groundwater up-welling flow under the out-of-phase scenario compresses the hyporheic zone extension during the flood event. Consequently, exfiltrating hyporheic fluxes under in-phase conditions increase twice as much as exfiltrating hyporheic fluxes under out-of-phase conditions. In contrast, the differences of exfiltrating hyporheic fluxes between in-phase and out-of-phase scenarios are marginal in response to the same flood event under losing conditions (Fig. 5d). Reasons will be explored in the following section.

This observation has potential implications on optimizing aquifer pumping schedule. Hypothetically, if the rising discharge is from an untreated wastewater discharge source, the timing of the groundwater table drawdown will significantly affect the spreading and mixing of pollutants in the sediment. At the moment of flood events, more pollutants will be carried into the sediment with a higher hyporheic exchange rate under a relatively low upwards-directed pressure of the groundwater than under a relatively high upwards-directed pressure. Therefore, the timing of the aquifer pumping can potentially amplify or reduce the dispersal of pollutants in the aquifer.

Modern regulating reservoirs are usually designed with enough storage capacities allowing planning of pumping schedules independent of user demand (Reca et al., 2014). A poorly designed pumping regime is detrimental to the biological and ecological functioning of the fluvial systems (Moore, 1999; Libera et al., 2017; Bredehoeft and Kendy, 2008). Consequently, careful selection of aquifer pumping schedules with considerations of both timing of flood and groundwater table dynamics are critical for water management agencies to minimize the environmental footprint of the withdrawal process.

## 4.2   Different Impacts of Groundwater on Hyporheic Exchange Under Gaining and Losing Systems

The timing of groundwater table drawdown has substantially different impacts on hyporheic exchange processes under gain-ing and losing conditions in different seasons. More specifically, under gaining conditions, the opposite phases of groundwater table fluctuations under in-phase and out-of-phase conditions induce an opposite fluctuation pattern of exfiltrating hyporheic fluxes in both winter and summer (the black and blue solid lines in Fig. 3e and 3f). However, under losing conditions the opposite fluctuation patterns of exfiltrating hyporheic fluxes under in-phase and out-of-phase conditions gradually disappear with increasing river temperatures from winter to summer (the black and blue solid lines in Fig. 5c and 5d). Differing from gaining conditions, under losing conditions, exfiltrating hyporheic fluxes in both in-phase and out-of-phase scenarios present an almost synchronized fluctuation pattern following the diel river temperature fluctuations in summer. These results indicate that under losing conditions, even though both river temperature and timing of groundwater table drawdown affect the phase of exfiltrating hyporheic flux fluctuations in winter when river temperatures are relatively low, river temperature, however,

plays a more dominant role in determining the phase of the hyporheic flux fluctuations in summer when river temperatures are relatively high. In other words, higher river temperature has larger impacts on the temporal variations of hyporheic exchange.

To better understand the causes of different hyporheic responses under gaining and losing conditions with relatively high river temperatures (i.e. in summer), snapshots of sediment temperature distributions on a summer afternoon are presented (Fig. 6). Under gaining conditions, areas affected by the river temperature are closely dependent on the hyporheic exchange processes (Fig. 6a). When hyporheic exchange rate is low, the river temperature has a negligible effect on the sediment hydraulic conductivity because the heat advection of upwelling groundwater is dominant. When hyporheic exchange rates are relatively high, hyporheic zones will extend deeper and wider in the sediment and river bank (Gomez-Velez et al., 2017; Wu et al., 2018). As a consequence, river temperature will have a larger impact on the sediment hydraulic conductivity. Under losing conditions, however, the sediment hydraulic conductivity is predominantly affected by the surface water heat advection and conduction (Fig. 6b).

With the temperature variation approximately from 0 °C to 30 °C, viscosity decreases by 45% and hydraulic conductivity increases by 220% (Wu et al., 2020). Therefore, in summer when river temperature is relatively high, the hydraulic conductivity is enhanced and becomes the main modulator for hyporheic exchange rate under losing condition. Compared with hydraulic conductivity, the effect of daily fluctuations of groundwater gradients becomes less important in determining the variability of hyporheic exchange processes. Consequently, the differences of exfiltrating hyporheic fluxes between in-phase and out-of-phase losing conditions disappear in summer. This also explains the different effects of the timing of groundwater table drawdowns during the same flood event on Jul 27 under gaining (Fig. 3f) and losing conditions (Fig. 5d). Unlike gaining conditions, under losing condition, the differences between flood-induced increases of exfiltrating hyporheic fluxes in in-phase and out-of-phase scenarios are negligible, because river temperatures have a more dominant role in determining the variability of hyporheic exchange fluxes under losing systems.

It is noteworthy that when river temperature is relatively high, the exfiltrating hyporheic fluxes under out-of-phase gaining condition fluctuate with a higher amplitude (Fig. 3f). This is because under gaining out-of-phase scenario, a lower groundwater table (also means lower groundwater upwelling fluxes) occurs in the afternoon when river temperature is relatively high. Both a low groundwater upward gradient and a high river temperature promote hyporheic exchange. Consequently, the exfiltrating hyporheic fluxes fluctuate with a higher amplitude under out-of-phase gaining conditions than under in-phase conditions.

When gradually reducing the groundwater fluctuation amplitudes, the crests of exfiltrating hyporheic fluxes under in-phase gaining groundwater scenario shift from the timing of river temperature troughs to river temperature peaks (Fig. 4b). This is another clear evidence that both diel river temperatures and groundwater daily fluctuations regulate the phases and amplitudes of hyporheic exchange fluxes: when the groundwater fluxes are small, the diel rhythm of hyporheic flux fluctuations is following the diel fluctuations of river temperature; whereas when the groundwater fluxes increase, the diel rhythm of hyporheic flux fluctuations is following the timing of groundwater level daily drawdown.

### 4.3 Groundwater Modifies Hyporheic Buffering Effects on Temperature

Temperature differences between river and exfiltrating hyporheic fluxes also demonstrate distinct patterns between gaining
and losing, in-phase and out-of-phase conditions. Under gaining conditions, the temperature differences display negative values
in winter periods and positive values in summer periods due to the mixing between surface water and groundwater (Fig. 7a).
In winter, the groundwater is often warmer than surface water; while in summer, the groundwater is often colder than surface
water. Therefore, temperature differences under gaining conditions demonstrate a clear seasonal fluctuations around zero.
Unlike gaining conditions, temperature differences under losing conditions have no clear seasonal fluctuations around the
value zero due to the limited mixing between regional groundwater and surface water.

The temperature differences between exfiltrating hyporheic fluxes between in-phase and out-of-phase gaining conditions are
directly related to the temporal variability of hyporheic exchange fluxes (Fig. 3e and 3f) and sediment temperature distribution.
As discussed above, the hyporheic exchange rate is higher under out-of-phase conditions than under in-phase conditions when
river temperatures are relatively high. As a result, the hyporheic zone has a larger extension and surface water can infiltrate
deeper into the sediment. Therefore, hyporheic zones have a larger cooling effect during high river temperature under out-of-
phase gaining conditions than under in-phase gaining conditions.

Spatial variability in river and sediment temperature may provide localized refugia against extreme thermal disturbances
for aquatic communities (Berman and Quinn, 1991). Loss of these refugia increases the risk for organisms living under unde-
sirable temperatures associated with diel temperature fluctuations and anthropogenic activities (Poole and Berman, 2001). In
the present study, we observe that the timing of daily groundwater table drawdown (i.e. in-phase or out-of-phase scenarios)
potentially affects the ability of hyporheic zones to act as temperature buffers that can sustain vital activities (i.e., survival,
growth and reproduction) for aquatic communities. Therefore, care must be taken in scheduling the pumping activities in order
to protect thermal heterogeneity across multiple spatial scales.

### 4.4 Groundwater Modifies Hyporheic Potential for Biogeochemical Reactions

Hyporheic potential for denitrification varies between gaining and losing, in-phase and out-of-phase conditions (Fig. 8).
$RSF_a$ displays substantial drops during flood events. This is because flood-induced hydraulic gradient increases at the sediment-
water interface drive more surface water into the sediment, and consequently accelerate hyporheic exchange rates. Increased
hyporheic exchange rates lead to a substantial decrease of the residence time in the hyporheic zone, creating conditions less
suitable for denitrification. Similarly, $RSF_a$ under gaining conditions is around three orders of magnitude higher than under
losing conditions due to the significantly longer residence time resulting from mixing between surface water and groundwater
under gaining conditions.

With groundwater gaining conditions, $RSF_a$ peaks at different time during a day under in-phase and out-of-phase scenarios,
indicating hyporheic denitrification potential can be regulated by adjusting the timing of daily groundwater table drawdowns.

With groundwater losing conditions, even though $\text{RSF}_a$ display peaks at different times during a day on a logarithmic scale under in-phase and out-of-phase scenarios, the actual differences of $\text{RSF}_a$ (in the scale of 10 to the power of $-5$) between in-phase and out-of-phase conditions are insignificant compared to gaining conditions (Fig. 8a and 8b). In conclusion, the timing of groundwater table drawdown is more important under gaining conditions than under losing conditions for denitrification reactions.

In Harvey et al. (2019), RSF was calculated based on mean annual hyporheic flux and river discharge without considerations of the temporal variability of the flow conditions and groundwater upwelling/downwelling. To be able to compare our results with those results, we also calculated mean RSF using mean river discharge and mean hyporheic fluxes. The calculated mean RSF is approximately -2.7 to -1.8 for gaining condition and -5.8 to -4.8 for losing condition, which roughly falls within the range of the mean RSF observed in Harvey et al. (2019). Under losing condition, the RSF is smaller than the values reported in Harvey et al. (2019), because losing conditions significantly reduce the denitrification potential as indicated in Fig. 8.

It's worth mentioning that the observations of $\text{RSF}_a$ are not limited to denitrification processes. For a different biogeochemical reaction, another characteristic time scale is applied instead of $\tau_{dn}$. Results presented in Fig. 8 will only be scaled by a different biogeochemical time scale for the reaction of interest. The relative variations of $\text{RSF}_a$ remain the same for other biogeochemical reactions.

The temperature-dependence for $\tau_{dn}$ is not considered, however we use both the $25^{th}$ and $75^{th}$ quantiles as the lower and upper ranges for calculating $\text{RSF}_a$, which mostly include the variations caused by the changing temperature as indicated in Zheng et al. (2016) where a roughly five-fold increase was observed in denitrification rates when temperature increased from $5°C$ to $35°C$.

The first term in $\text{RSF}_a$ ($q_{HZ}/Q$) describing the proportion of river discharge passing through the hyporheic zone per unit bedform area can be used to quantify the connectivity between river and hyporheic zone (Harvey et al., 2019). This connectivity underpins many ecosystem processes and important reactions that take place in close contact with biogeochemical reactive sediments (Boulton, 2007; Ward et al., 2000; Malard et al., 2002; Roley et al., 2012). Maintaining a good hydrological connectivity is therefore crucial. Under the same river discharge rates ($Q$), hyporheic exchange rates ($q_{HZ}$) are higher when groundwater drawdown is out-of-phase to diel river temperature fluctuations than in-phase. Consequently, the hydrological connectivity is higher in an out-of-phase scenario. The temperature differences between river and exfiltrating hyporheic fluxes with in-phase and out-of-phase groundwater table drawdown also proves this finding (Fig. 7). Hydrological connectivity is higher in out-of-phase groundwater table fluctuation scenarios than in in-phase scenarios, making the hyporheic zone a better thermal buffer under out-of-phase scenarios.

## 4.5  Study limitations

The aim of the present study is not to simulate hyporheic exchange processes with perfect details, but rather to gain mechanistic understanding of hyporheic responses to varying groundwater table fluctuation patterns. Therefore, simplifications are

made to allow for an efficient and reasonably correct representation of hyporheic exchange processes. Detailed simplifications and limitations on model dimensionality, geomorphological settings, and boundary conditions are critically reviewed in previous studies on which the development of current method is based (Wu et al., 2018, 2020). In the following, only simplifications that are most relevant to the present study are discussed.

Groundwater fluxes are simplified as prescribed upward or downward fluxes. Daily groundwater table drawdowns are represented by sinusoidal curves with different phases and amplitudes representing different timing of groundwater table drawdowns and strength of groundwater upwelling or downwelling, respectively (Fig. 2). However, the direction and magnitude of groundwater flow is a response to the head difference between river stage and riparian water table elevation, as well as sediment properties. An important process that cannot be represented by using prescribed groundwater fluxes is the impact of river tem-
perature as a major factor contributing to reduced afternoon river discharge. High river temperature in the afternoon results in a high hydraulic conductivity which contributes to increased losing fluxes and consequently a reduced afternoon river discharge (Constantz et al., 1994). However, increasing of losing fluxes due to higher river temperature in the afternoon cannot be captured using a prescribed groundwater flux time series. Apart from changing sediment hydraulic conductivity, there are a myriad of other factors affecting groundwater table fluctuations. For instance, a flood event may change the head difference between
river stage and riparian water table elevation, and eventually leads to changes in the direction and magnitude of groundwater flow (Todd and Mays, 2005; Lewandowski et al., 2009). The head difference may change from negative to positive, resulting in a switch of groundwater gaining to losing condition. However, these changes cannot be represented by using a prescribed groundwater flux time series. Groundwater table as a direct response to the head difference between the adjacent aquifer and the river stage is hence suggested for future hyporheic modeling in order to account for the hyporheic dynamics introduced by
natural groundwater table fluctuations.

    Additionally, in the present study the surface water flow is assumed as an independent system that is not affected by groundwater flows. This simplification can only be used when groundwater discharge or recharge is significantly smaller than river discharge. In our case, the groundwater discharge or recharge is at least 4 orders of magnitude lower than river discharge. Therefore, this simplification has limited impact on the results. The noticeable difference in the magnitude between ground-
water discharge/recharge and river discharge also emphasizes the finding that even small groundwater fluxes may have a pronounced influence on the hyporheic zone.

    morphological setting of the model is dune with aspect ratio of 0.1 under subcritical flow conditions with a Froude number around 0.39 (Dingman, 2009; Bridge, 2009). The geological setting has been simplified as homogeneous and isotropic porous media. Even though the sediment in nature can rarely be homogeneous and isotropic, this simplification is necessary
for improving computational efficiency without defeating the objective of identifying the interactions among river discharge, temperature and groundwater dynamics.

## 5 Conclusions

Groundwater table dynamics substantially modulate hyporheic exchange processes. Daily groundwater withdrawal causes additional variability of hyporheic exchange besides the variability induced by the diel river temperature changes. However, the variability induced by daily groundwater table drawdown is not necessarily an addition to the fluctuations induced by the diel river temperature changes. More specifically, groundwater flow fluctuations that are out-of-phase to diel river temperature fluctuations are likely to promote hyporheic exchange to a larger extent than groundwater flow fluctuations that are in-phase to diel river temperature fluctuations. Even though both groundwater table fluctuations and diel river temperature fluctuations affect hyporheic exchange dynamics, under the same discharge condition, river temperature has a more dominant role in determining hyporheic exchange variability under losing conditions than under gaining conditions. This is because under gaining conditions, heat advection of upwelling groundwater is more dominant; under losing conditions heat advection and conduction of surface water is more dominant in hyporheic zone's heat exchange.

The timing of groundwater table drawdown modifies the rates of hyporheic exchange, and as a result the mixing and spreading of pollutants in the aquifer. Pumping activities should be avoided during flood events in order to ensure minimal contaminant uptake. Additionally, the timing of groundwater table drawdown also affects the hyporheic zone's ability to act as a temperature buffer that protects aquatic communities from thermal extremes. Although not as significant as the effect of flood events, hyporheic denitrification potential (and potentially for other biogeochemical reactions) is also changing following the groundwater table drawdown. Therefore, careful considerations must be taken when planning aquifer pumping schedules in order to minimize negative environmental impacts.

*Data availability.* All data required to reproduce the figures in this paper is available on the database of Leibniz-Institute of Freshwater Ecology and Inland Fisheries (https://www.igb-berlin.de/freshwater-research-and-environmental-database)

*Author contributions.* LW, JGV, and JL designed the study layout with the feedback from SK, AW, TS, and GN. LW performed simulations, analyzed the data, and prepared the manuscript with the contributions from JGV, AW, TS, and JL. The manuscript was reviewed by JGV, SK, AW, TS, GN, and JL.

*Competing interests.* The authors declare that they have no conflict of interest.

*Acknowledgements.* This study has received funding from the European Union's Horizon 2020 research and innovation programme under Marie Sklodowska-Curie grant agreement No. 641939 (HypoTRAIN) and No. 734317 (HiFreq). Additional funding was granted by the German Research Foundation (DFG) for the Research Training Group under No.GRK 2032/1 (Urban Water Interfaces). J.D. Gomez-

Velez is funded by the U.S. National Science Foundation (award EAR 1830172) and the U.S. Department of Energy, Office of Biological and Environmental Research (BER), as part of BER's Subsurface Biogeochemistry Research Program (SBR). This contribution originates from the SBR Scientific Focus Area (SFA) at the Pacific Northwest National Laboratory (PNNL). T. Singh is partly supported by the German Research Foundation under the grant WO671/11-1. We acknowledge the comments and suggestions from editor Christian Stamm and three anonymous reviewers who helped improve this paper significantly.

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
