# Peer review of "How does daily groundwater table drawdown affect the diel rhythm of hyporheic exchange?"

_Hydrology and Earth System Sciences, 2020_

## Referee Comment (RC1) · Anonymous Referee #1 · 30 Aug 2020

Wu et al, 2020 Review

How does daily groundwater table drawdown affect the diel rhythm of hyporheic exchange? Initial Comments The authors of this paper use USGS gauge data with diel fluctuations in discharge and river temperature to model hyporheic exchange rates in order to better understand how daily groundwater table fluctuations change hyporheic exchange rates in gaining, losing, or neutral streams. The authors use complex modeling to show how in-phase or out-of-phase daily groundwater table drawdown can influence hyporheic exchange rates. The model created for this paper makes hard assumptions about river morphology, network position, and sediment characteristics to step back and look at daily groundwater table dynamics conceptually. While much of the paper is modeling hyporheic exchange the authors also ask how diel ground-

water table fluctuations and river temperature impact residence time for denitrification potential and thermal refugia for aquatic species. The authors conclude that groundwater table dynamics modulate hyporheic exchange process differently than diel river temperature. When diel groundwater table drawdown is out-of-phase with river temperature hyporheic exchange is greater than when in-phase. Under gaining conditions upwelling groundwater buffers diel river temperature and increases hyporheic exchange rates. Under losing conditions surface water temperature penetrates deeper into the hyporheic zone and decreases hyporheic exchange rates. The authors do a good job in the modeling and data analysis sections of this research yet need to make the objectives of this paper clearer to support findings of this paper.

Specific Comments 1. The objective statement of this paper is not well defined. After a good introduction, the last paragraph is lacking in clarity as to what this paper is about. Suggestion for the authors to use language like: "In the present study, we aim to quantify the impact of groundwater withdrawal on hyporheic exchange processes at the daily scale as well as better understand impacts on potential denitrification and thermal buffering". Then move on to how this paper accomplished the objectives. "To investigate these objectives, we built a complex model that. . .." This will also help guide the reader towards the start of the methods section. 2. The connection from the modeling to RSF and thermal refugia for aquatic species is weak. It feels like the nutrient processing and ecosystem services provided by hyporheic exchange are tossed into this paper to try to broaden the scope of the paper. I suggest that the authors leave nutrient processing to the discussion section rather than a main objective of this paper. Much of the paper does good modeling of hyporheic exchange rates and that should be the focus. There is also some confusion in if this paper wants to just focus on denitrification or RSF and this distinction needs to be clear to the audience. The authors also provide no hard numbers as to how RSF was applied to their model. The Gomes-Velez (2016) paper provides a range of RSF for stream orders 1-12 and how RSF varies throughout stream orders. The authors fail to mention what RSF values were chosen amongst that range. While the result of the RSF analysis is interesting, the explanation as to what

this mean ecologically is missing. 3. Hyporheic connectivity is not discussed or mentioned in this paper. How does connectivity change during these diel fluctuations or during storms? How connected the hyporheic zone is could impact the thermal buffering capacity. A short paragraph on this topic should be added. 4. The is also confusion as to what a groundwater table drawdown means. The diel groundwater fluctuations presented here are due to plant uptake, yet the authors also mention groundwater pumping. The introduction paragraph (Lines 64-70) sets up the pumping problem well but does not mention plants. The discussion section does not discuss the pumping problem well enough to support the management implications in the conclusion. The implications for poorly designed pumping schedules are huge given your data during the flood event! 5. The conclusion is also weak and does not drive home the answers found from the objective statement. The closing sentence is subjective and needs to be reworded: "Our data show that hyporheic exchange rates in a gaining river increase significantly during storm events. When combined with an in-phase diel groundwater table fluctuation, hyporheic exchange rates are higher than an out-of-phase fluctuation (Fig 3f storm vs. Fig 5f storm). Anthropogenic aquifer pumping schedules should be out of phase with diel river temperature to ensure minimal contaminant uptake". RSF or denitrification also needs to me worded stronger here. 6. Transitional sentences between paragraphs and sections need to be stronger making it hard for the reader to follow

Technical Comments  c Abstract ok o The phrasing of groundwater withdrawal makes it sound like there is anthropogenic influence. You do not specifically look at this so I would keep it to the discussion section  c Line 14, I would turn this first sentence in a strict definition of the hyporheic zones o Something like hyporheic zones are transitional areas between surface water and groundwater environments that often exhibit marked physical, chemical, and biological gradients that drive the exchanges of water flow, energy, solute and microorganisms between surface and subsurface regions. o This will help focus the readers the research in this paper  c Line 18, what makes researching spatiotemporal variability of hyporheic exchange key to water

resources management? Provide a reference • Line 19, what and how is it key to ecosystem restoration • Line 23, change to factors influencing the hydraulic. . .. • Line 26, change language. Make this more clear • Entire second paragraph needs to be worded better • Line 43 – Good sentence here • Figure 1 o Groundwater table A and B separation is confusing to the eye o Do these relate to either the gaining or losing condition o Suggestion to color the lines differently o Remove the tree image or add more. Suggestion to use a tree silhouette. • Line 45, reference needed for 1st sentence • Line 58, Wu et al. observed. . .. • Line 71, This entire paragraph needs to be stronger • Transition from objective statements to modeling section is poor o Ideas for objective statements  Stronger, need to be more focused. This paragraph is short and weak when it should be the strongest hit of the paper • In the present study, we aim to quantify the impact of groundwater withdrawal on hyporheic exchange processes at the daily scale as well as better understanding river temperature impacts on potential denitrification and thermal buffering. o Modeling transition  Use the last paragraph to transition to the modeling • This is poor • Line 80, need a transition sentence to connect to the aims • Line 84, need reference for COMSOL method and mesh-independent. • Figure 2 o Good conceptual figure • Figure 3 o Say that discharge is not to scale, rather than not labeled. Or that you are using it for visual aid and not to scale • Line 214, you say only in-phase results are shown but Figures 3 and 5 show out of phase results • Figure 4 o I don't like the positioning of Figure 4 but don't know if you have control over this or the journal does. It looks odd to have a figure showing gaining conditions in the 3.1.2 under Losing Conditions section of the paper • Figure 5 o Caption says discharge is not labeled when it is in Fig 5c and Fig 5d o I think you may mean that discharge is not to scale in 5e and 5f • Line 260, please state the values you used for you models or at least a range of values • Figure 6 o I'm not sure how necessary figure 6 is in this paper. While I like the figure, I believe you could and do explain this information in the text. o This could help you shorten the paper o You could slow spice this up by clipping a few of these snapshots together and then playing them in a .gif over

the course of a storm so you could see the variations in the losing condition sections of the figure o • Figure 7 o Same weird out of place figure placement o I like this figure. It tells me clearly that gaining in-phase hyporheic zones have less variable temperature from the constant upwelling of groundwater o Get rid of the underscore in gaining in-phase, keep it consistent with the figures above. Same goes for the color scheme if possible • Line 260 o Gomez-velez et al 2015 reports RSF over entire river networks. How you are you implementing these findings into this new model? The also include river bedform information and this paper assumes uniform sediment. So please list what metrics you are using from this Gomez-velez paper. What are the quantiles??? • Figure 8 o Under loosing conditions reaction significance time is 3 orders of magnitude less than gaining conditions o This figure indicate that the RSF can vary by ∼1 order of magnitude over the course of the day. While the difference between gaining and loosing conditions is and interesting result. How do you justify this with the range of stream orders, sediment size, and hydraulic conductivity show in the Gomez-Veles papers? o Are you using the stream order of the USGS gauge you gather the data from? If so report these information and explain this process in the text • Discussion • Line 267, Water table drawdowns coupled with hydraulic gradient changes through temperature contribute to enhanced diel fluctuations of exfiltrating hyporheic fluxes • Line 269, Under the neutral condition • Line 272, 269 – o You only reference figure 3 here which is the gaining condition, should you also mention figure 5 the loosing condition? o Or be more specific in the text • Paragraph on Line 285 o I agree with what you are saying o Don't pump an aquifer during a storm because the drawdown could pull pollutants into the hyporheic zone o Could you provide an example of a usgs site that has daily drawdowns from groundwater pumping like the ones shown in this paper from the plants? o This may be a hard reach but could have important management implications • Line 307, could you use your data (from figure 6 maybe) to show this? o Upwelling keeps warm surface water from connecting to HZ • Therefore, in summer when river temperature is relatively high, the hydraulic conductivity is enhanced and becomes the main modulator for

hyporheic exchange rate under losing condition. o Change the therefore language. The authors use this word a lot • Combine the paragraphs between Lines 310 and 320 • Line 343, Therefore, hyporheic zones have a larger cooling effect during high river temperature under out-of-phase gaining conditions than under in-phase conditions (under gaining conditions) • Too many conditions maybe think of different wording for in-phase and out-of-phase (conditions) • Loosing conditions speeds up residence time (RSF = reaction scale factor) • Gaining conditions slows down residence time and allows mixing of GW and SW • In conclusion, the timing 365 of groundwater table drawdown is more important under gaining conditions than under losing conditions for denitrification reactions. • Line 668 – could you mention this fact earlier in the paper, so the reader is not thinking about denitrification the entire time? • Study limitations? o What about connectivity? A reference to some of this great work would be nice to see in this paper • Conclusion o Not strong enough or long enough o Need more space and references to specific aquatic community impacts and groundwater table diel drawdown.

Please also note the supplement to this comment:
https://hess.copernicus.org/preprints/hess-2020-288/hess-2020-288-RC1-supplement.pdf

---

## Author Comment (AC1) · 22 Sep 2020

**Response to Comments from Referee #1**

**Initial Comments**

The authors of this paper use USGS gauge data with diel fluctuations in discharge and river temperature to model hyporheic exchange rates in order to better understand how daily groundwater table fluctuations change hyporheic exchange rates in gaining, losing, or neutral streams. The authors use complex modeling to show how in-phase or out-of-phase daily groundwater table drawdown can influence hyporheic exchange rates. The model created for this paper makes hard assumptions about river morphology, network position, and sediment characteristics to step back and look at daily groundwater table dynamics conceptually. While much of the paper is modeling hyporheic exchange the authors also ask how diel groundwater table fluctuations and river temperature impact residence time for denitrification potential and thermal refugia for aquatic species. The authors conclude that groundwater table dynamics modulate hyporheic exchange process differently than diel river temperature. When diel groundwater table drawdown is out-of-phase with river temperature hyporheic exchange is greater than when in-phase. Under gaining conditions upwelling groundwater buffers diel river temperature and increases hyporheic exchange rates. Under losing conditions surface water temperature penetrates deeper into the hyporheic zone and decreases hyporheic exchange rates. The authors do a good job in the modeling and data analysis sections of this research yet need to make the objectives of this paper clearer to support the data presented in this paper.

> **Response:** Thank you for the positive comment on the modeling and data analysis, and also for the insightful suggestions on improving the manuscript. The point-by-point reply to the comments is given below. Changes that will be made in the manuscript after the online discussion are indicated by *underlined text in italic*. Line numbers in this response refer to the numbers in the original manuscript. By responding to the following comments, we incorporated the changes to clarify the objectives, model assumptions, and to improve the structure for a better readability.

**Specific Comments**

1. The objective statement of this paper is not well defined. After a good introduction, the last paragraph is lacking in clarity as to what this paper is about. Suggestion for the authors to use language like: "In the present study, we aim to quantify the impact of groundwater withdrawal on hyporheic exchange processes at the daily scale as well as better understand impacts on potential denitrification and thermal buffering". Then move on to how this paper accomplished the objectives. "To investigate these objectives we built a complex model that…." This will also help guide the reader towards the start of the methods section.

   **Response:** Thank you for the good suggestion. To better present the objectives, we modified the last paragraph of the introduction (from Line 71-75) in the following way:

   *"In the present study, we aim to quantify the impact of river temperature fluctuations and groundwater withdrawal on hyporheic exchange processes at the daily scale, as well as to better understand implications on hyporheic zone's potential for denitrification and thermal buffering. With these objectives in mind, different groundwater scenarios corresponding to different timings of groundwater withdrawal under gaining and losing conditions are applied in a physically based hyporheic flow and heat transport model."*

2. The connection from the modeling to RSF and thermal refugia for aquatic species is weak. It feels like the nutrient processing and ecosystem services provided by hyporheic exchange are tossed into this paper to try to broaden the scope of the paper. I suggest that the authors leave nutrient processing to the discussion section rather than a main objective of this paper. Much of the paper does good modeling of hyporheic exchange rates and that should be the focus. There is also some confusion in if this paper wants to just focus on denitrification or RSF and this distinction needs to be clear to the audience. The authors also provide no hard numbers as to how RSF was applied to their model. The Gomes-Velez (2016) paper provides a range of RSF for stream orders 1-12 and how RSF varies throughout stream orders. The authors fail to mention what RSF values were chosen

amongst that range. While the result of the RSF analysis is interesting, the explanation as to what this mean ecologically is missing.

**Response:** To better address this comment, we will answer the three sub-comments in the following order:

1) How does RSF calculated? Were RSF values chosen from Gomez-Velez (2016)?

The RSF values were not chosen from Gomez-Velez (2016). They were calculated under the specific flow and sediment characteristics in the present study by Eq 8 which was first introduced by Harvey et al. (2013):

$$RSF_a = \frac{q_{HZ}}{Q} \cdot \frac{\tau_{HZ}}{\tau_{dn}}$$ (Eq. 8)

where $Q$ is the river discharge, $q_{HZ}$ is the exfiltrating hyporheic fluxes calculated with Eq 1-3, $\tau_{HZ}$ is the mean residence time of hyporheic flow calculated with Eq 6, $\tau_{dn}$ is the characteristic time scale for denitrification determined based on Gomez-Velez and Harvey (2014) and Gomez-Velez et al. (2015).

To better present the values of $\tau_{dn}$, we will add the following figure in the supplementary information to show the quantiles of the characteristic time scales for denitrification.

[Figure]

*Figure S1: Box plot of the characteristic time scale for denitrification (log₁₀[h]). The 25ᵗʰ quantile is 0.38, the 50ᵗʰ quantile is 0.87, and the 75ᵗʰ*

*quantile is 1.28. (Taken from Gomez-Velez and Harvey (2014) and Gomez-Velez et al. (2015))*

2) Does the paper focus on denitrification or RSF? What are the differences?

We calculated the reaction significance factors for denitrification with equation 8. However, the interpretations of results shown in Figure 8 are not limited to denitrification processes. For a different biogeochemical reaction, another characteristic time scale is applied instead of $\tau_{dn}$. Results presented in Figure 8 will only be scaled by a different biogeochemical time scale for the reaction of interest. The relative variations of RSF remain the same for other biogeochemical reactions (line 367-370).

3) Are nutrient processing and ecosystem services main focuses on this paper?

We partly agree with the reviewer. The ecosystem service is not the main focus of this paper. The impact on ecosystem was not quantified but only discussed in Discussion to show the impact of timing of groundwater table drawdown on the hyporheic zone's function as thermal buffers for aquatic communities qualitatively. However, the nutrient processing as we presented for denitrification was quantified with equation 8 and the results are presented in Figure 8. We think the results have clearly demonstrated the different impacts of groundwater table fluctuation on reaction potentials under gaining and losing conditions, which is a worthwhile message for readers who are interested in exploring biogeochemical reactions under different groundwater conditions. Therefore, we would like to keep the quantifications of reaction potential in the main objective.

3. Hyporheic connectivity is not discussed or mentioned in this paper. How does connectivity change during these diel fluctuations or during storms? How connected the hyporheic zone is could impact the thermal buffering capacity. A short paragraph on this topic should be added.

**Response:** This is a good point. We added a short paragraph about hyporheic connectivity by the end of discussion in line 371:

*The first term in $RSF_a$ ($\frac{q_{HZ}}{Q}$) describing the proportion of the river discharge passing through the hyporheic zone per unit bedform area can be used to quantify the connectivity between river and hyporheic zone (Harvey et al., 2019). This connectivity underpins many ecosystem processes and important reactions that take place in close contact with biogeochemical reactive sediments (Boulton, 2007; Ward et al., 2000; Malard et al., 2002; Roley et al., 2012). Maintaining a good hydrological connectivity is therefore crucial. Under the same river discharge rates (Q), hyporheic exchange rates ($q_{HZ}$) are higher when groundwater drawdown is in an out-of-phase pace to diel river temperature fluctuations than in an in-phase pace. Consequently, the hydrological connectivity is higher in a groundwater out-of-phase scenario. The temperature differences between river and exfiltrating hyporheic fluxes with in-phase and out-of-phase groundwater table drawdown also proves this finding (Fig. 7). Hydrological connectivity is higher in out-of-phase groundwater table drawdown scenarios than in in-phase scenarios, making hyporheic zone a better thermal buffer.*

*References added:*
- *Boulton, A. J. (2007). Hyporheic rehabilitation in rivers: restoring vertical connectivity. Freshwater Biology, 52(4), 632-650.*
- *Harvey, J., Gomez‐Velez, J., Schmadel, N., Scott, D., Boyer, E., Alexander, R., ... & Moore, R. (2019). How hydrologic connectivity regulates water quality in river corridors. JAWRA Journal of the American Water Resources Association, 55(2), 369-381.*
- *Malard, F., Tockner, K., DOLE‐OLIVIER, M. J., & Ward, J. V. (2002). A landscape perspective of surface‐subsurface hydrological exchanges in river corridors. Freshwater Biology, 47(4), 621-640.*
- *Roley, S. S., Tank, J. L., & Williams, M. A. (2012). Hydrologic connectivity increases denitrification in the hyporheic zone and restored floodplains of*

*an agricultural stream. Journal of Geophysical Research: Biogeosciences, 117(G3).*

- *Stanford, J. A., & Ward, J. V. (1993). An ecosystem perspective of alluvial rivers: connectivity and the hyporheic corridor. Journal of the North American Benthological Society, 12(1), 48-60.*
- *Ward, J. V., Malard, F., Stanford, J. A., & Gonser, T. (2000). Interstitial aquatic fauna of shallow unconsolidated sediments, particularly hyporheic biotopes. In 'Subterranean Ecosystems'.(Eds H. Wilkens, DC Culver and WF Humphreys.) pp. 41–58.*

4. The is also confusion as to what a groundwater table drawdown means. The diel groundwater fluctuations presented here are due to plant uptake, yet the authors also mention groundwater pumping. The introduction paragraph (Lines 64-70) sets up the pumping problem well but does not mention plants. The discussion section does not discuss the pumping problem well enough to support the management implications in the conclusion. The implications for poorly designed pumping schedules are huge given your data during the flood event!

> **Response:** Thank you for the comments. As introduced in line 46-48, both the phreatophytes water-use and anthropogenic pumping can cause groundwater table drawdown at a daily scale. Therefore, the daily groundwater table drawdowns in the present study were conceptualized as sinusoidal curves with varying amplitudes and phases.
>
> We also understand the reviewer's confusion on line 64-70. To clarify the research set up, the description of pumping problem will be removed from introduction. This issue instead will only be discussed in Discussion to present the implications of our results on pumping management. A short paragraph will be added in line 288:

*Modern regulating reservoirs are usually designed with enough storage capacities allowing planning of pumping schedules independent of user demand (Reca et al., 2014). A poorly designed pumping regime is detrimental to the biological and ecological functioning of the fluvial systems (Moore, 1999; Libera et al., 2017; Bredehoeft and Kendy, 2008). Consequently, careful selection of aquifer pumping schedules with considerations of both timing of flood and groundwater table dynamics are critical for water management agencies to minimize the environmental footprint of the withdrawal process.*

5. The conclusion is also weak and doesn't drive home the answers found from the objective statement. The closing sentence is subjective and needs to be reworded: "Our data show that hyporheic exchange rates in a gaining river increase significantly during storm events. When combined with an in-phase diel groundwater table fluctuation, hyporheic exchange rates are higher than an out-of-phase fluctuation (Fig 3f storm vs. Fig 5f storm). Anthropogenic aquifer pumping schedules should be out of phase with diel river temperature to ensure minimal contaminant uptake". RSF or denitrification also needs to me worded stronger here.

   **Response:** Thank you for the comment. The conclusion has been rephrased. Please find it in the response to the last comment on Conclusion.

6. Transitional sentences between paragraphs and sections need to be stronger making it hard for the reader to follow

   **Response:** Thank you for the comment. We rephrased last paragraph of Introduction and entire Conclusions to improve the connections between sections. Please refer to responses to comments on line 71 and conclusions below.

**Technical Comments**

- Abstract ok
    - The phrasing of groundwater withdrawal makes it sound like there is anthropogenic influence. You do not specifically look at this so I would keep it to the discussion section

**Response:** Done as suggested. Groundwater withdrawal is replaced by groundwater level drawdown to reflect the groundwater level fluctuation in a more general way. The text now reads:

*The timing of groundwater table drawdown has a direct influence on hyporheic exchange rates and hyporheic buffering capacity on thermal disturbances.*

- Line 14, I would turn this first sentence in a strict definition of the hyporheic zones
  - Something like hyporheic zones are transitional areas between surface water and groundwater environments that often exhibit marked physical, chemical, and biological gradients that drive the exchanges of water flow, energy, solute and microorganisms between surface and subsurface regions.
  - This will help focus the readers the research in this paper

    **Response:** Thank you for the suggestion. We modified the sentence as suggested in line 14:

    *Hyporheic zones are transitional areas between surface water and groundwater environments, which often exhibit marked physical, chemical, and biological gradients that drive the exchanges of water flow, energy, solute and microorganisms between surface and subsurface regions.*

- Line 18, what makes researching spatiotemporal variability of hyporheic exchange key to water resources management? Provide a reference

    **Response:** Thank you for the question. The important role hyporheic zone playing in connecting surface and subsurface water environments as outlined in line 14-17 justified the necessity of understanding spatiotemporal variability of hyporheic exchange for water resources management. Here we added the following reference:

*Lewandowski, J., Arnon, S., Banks, E., Batelaan, O., Betterle, A., Broecker, T., ... & Gomez-Velez, J.: Is the hyporheic zone relevant beyond the scientific community? Water, 11(11), 2230, 2019.*

- Line 19, what and how is it key to ecosystem restoration

  **Response:** Hyporheic zone's effects on ecosystem restoration were not introduced with details, because as the reviewer suggested in the second specific comment it is not the main focus. Here we added a new reference of *Lewandowski et al., 2019* (listed in the comment on line 18 above) to help readers to find the relevant information.

- Line 23, change to factors influencing the hydraulic….

  **Response:** Done as suggested.

- Line 26, change language. Make this more clear

  **Response:** Please refer to the next comment below.

- Entire second paragraph needs to be worded better

  **Response:** Entire second paragraph is re-worded as below:

  *Hydrological drivers and modulators of time-varying hyporheic exchange processes have been extensively studied in the last decade. The hydraulic gradient as the main driver of hyporheic exchange processes is changing along the sediment-water interface, determining (1) the spatiotemporal variability of hyporheic zone extents and (2) characteristic time scales of hyporheic exchange (Boano et al., 2013; Ward et al., 2017; Gomez-Velez et al., 2017). Factors influencing the hydraulic gradient at the sediment-water interface include channel flow (Trauth and Fleckenstein, 2017; Grant et al., 2018; Broecker et al., 2018), geomorphological settings (Tonina and Buffington, 2011; Schmadel et al., 2016; Singh et al., 2019), and regional groundwater flow (Nützmann et al., 2014; Malzone et al., 2016; Wu et al., 2018). Sediment and fluid properties do not drive hyporheic exchange, but they modulate hyporheic exchange substantially:*

*sediment heterogeneity can alter hyporheic flow paths and residence time distributions, creating hot spots for biogeochemical transformations (Sawyer and Cardenas, 2009; Gomez-Velez et al., 2014; Pescimoro et al., 2019); fluid properties, i.e., density and viscosity, are functions of temperature and directly influence the hydraulic conductivity, thus hyporheic flow. Consequently, river temperature variability (i.e., diel and seasonal river temperature fluctuations) induces significant changes of hyporheic exchange processes (Cardenas and Wilson, 2007a). The spatiotemporal variability of the drivers and modulators eventually results in dynamic hyporheic exchange processes. Among these drivers and modulators, the combined effects of regional groundwater flow and river temperature on dynamic hyporheic exchanges are comparably understudied.*

- Line 43 – Good sentence here
  **Response:** Thank you!

- Figure 1
  o Groundwater table A and B separation is confusing to the eye
  **Response:** To better separate groundwater table A and B, we will color differently for groundwater table A and B.

  o Do these relate to either the gaining or losing condition
  **Response:** Yes, groundwater table A refers to gaining condition where the groundwater table is higher than river stage; groundwater table B refers to losing condition where groundwater table is lower than river stage,

  o Suggestion to color the lines differently
  **Response:** Thank you for the suggestion. We will do as suggested.

  o Remove the tree image or add more. Suggestion to use a tree silhouette.
  **Response:** We will use a tree silhouette as suggested.

- Line 45, reference needed for 1st sentence

**Response:** Done as suggested. The following reference will be added:

*Todd, D. K. and Mays, L. W.: Groundwater hydrology edition, Welly Inte, 2005.*

- Line 58, Wu et al. observed….

    **Response:** Done as suggested.

- Line 71, This entire paragraph needs to be stronger

    **Response:** This paragraph is rephrased. Please refer to the first specific comment and the next comment below.

- Transition from objective statements to modeling section is poor
    - Ideas for objective statements
        - Stronger, need to be more focused. This paragraph is short and weak when it should be the strongest hit of the paper
            - In the present study, we aim to quantify the impact of groundwater withdrawal on hyporheic exchange processes at the daily scale as well as better understanding river temperature impacts on potential denitrification and thermal buffering.

    **Response:** Thank you for the suggestion. We have modified as below:

    *In the present study, we aim to quantify the impact of river temperature fluctuations and groundwater table drawdown on hyporheic exchange processes at the daily scale, as well as to better understand implications on hyporheic zone's potential for denitrification and thermal buffering.*

    - Modeling transition
        - Use the last paragraph to transition to the modeling
            - This is poor

    **Response:** Thank you for the suggestion. The following sentences are added to act as a transition to the modeling section from line 73:

*With these objectives in mind, different groundwater scenarios corresponding to different timings of groundwater table drawdown under gaining and losing conditions are applied in a physically based hyporheic flow and heat transport model. Hyporheic exchange rates, temperature distribution and denitrification efficiency are quantified to assess the impacts of river temperature and groundwater level fluctuations on hyporheic exchange processes.*

- Line 80, need a transition sentence to connect to the aims

  **Response:** The following sentence is modified in line 80 to connect to the aims:

  *To understand the hyporheic exchange in response to changing river discharge, temperature and groundwater table fluctuations, a two-dimensional conceptualization is proposed based on Wu et al. (2018) and Wu et al. (2020) (Fig. 2a).*

- Line 84, need reference for COMSOL method and mesh-independent.

  **Response:** The COMSOL model was developed based on Wu et al. (2018) and Wu et al. (2020) (Fig. 2a) as mention in line 81.

- Figure 2
  - Good conceptual figure

    **Response:** Thank you!

- Figure 3
  - Say that discharge is not to scale, rather than not labeled. Or that you are using it for visual aid and not to scale

    **Response:** Thank you for the suggestion. The figure caption is modified as below:

    *For figure clarity, discharge is not scaled in e and f, and used only for visual aid.*

- Line 214, you say only in-phase results are shown but Figures 3 and 5 show out of phase results

  **Response:** Thank you for the comment! Effects of groundwater table fluctuation amplitudes on dynamic hyporheic responses are only explored under in-phase scenarios, because under out-of-phase scenarios, fluctuations of exfiltrating hyporheic fluxes are almost always in the same phase with the diel river temperature fluctuations. Therefore, unlike in-phase scenarios, the phase shifts due to reduced amplitudes in groundwater table fluctuation are not observed. Reduced amplitudes in groundwater table fluctuation under out-of-phase scenarios only contribute to reduced amplitudes in exfiltrating hyporheic flux fluctuations. Based on these reasons which are also stated in line 210-214, only results in in-phase scenarios are presented in figure 4.

  To clarify in the text, the following sentence is modified in line 214 as below:

  *For simplicity, only results in in-phase scenarios are presented in Fig. 4.*

- Figure 4
  - I don't like the positioning of Figure 4 but don't know if you have control over this or the journal does. It looks odd to have a figure showing gaining conditions in the 3.1.2 under Losing Conditions section of the paper

    **Response:** Thank you for the suggestion. We will fix the position of the figures.

- Figure 5
  - Caption says discharge is not labeled when it is in Fig 5c and Fig 5d
  - I think you may mean that discharge is not to scale in 5e and 5f

    **Response:** The caption is correct in referring to Fig 5c and Fig 5d. There are no fig 5e and 5f.

- Line 260, please state the values you used for you models or at least a range of values

**Response:** These values are stated in response to the second specific comment and presented in figure S1.

- Figure 6
  - I'm not sure how necessary figure 6 is in this paper. While I like the figure, I believe you could and do explain this information in the text.
  - This could help you shorten the paper
  - You could slow spice this up by clipping a few of these snapshots together and then playing them in a .gif over the course of a storm so you could see the variations in the losing condition sections of the figure

    **Response:** Thank you for this comment. Figure 6 conveyed an important message that the heat distribution is significantly different in gaining and losing groundwater systems under the same hydrological and climate condition. Although we could explain this information in the text, this figure has the direct visual explanation of this key point, which could help those readers who skip the text and only scan the figures to capture this important point. As the reviewer suggested, we will also include a gif figure showing the animation of the heat distribution along the course of changing discharge conditions.

- Figure 7
  - Same weird out of place figure placement
  - I like this figure. It tells me clearly that gaining in-phase hyporheic zones have less variable temperature from the constant upwelling of groundwater
  - Get rid of the underscore in gaining in-phase, keep it consistent with the figures above. Same goes for the color scheme if possible

    **Response:** Thank you for the suggestion. We will fix the position of the figure and remove the underscore in the figure legends.

- Line 260

o Gomez-velez et al 2015 reports RSF over entire river networks. How you are you implementing these findings into this new model? The also include river bedform information and this paper assumes uniform sediment. So please list what metrics you are using from this Gomez-velez paper. What are the quantiles???

**Response:** We have addressed this comment in the second specific comment.

- Figure 8
  - o Under loosing conditions reaction significance time is 3 orders of magnitude less than gaining conditions
  - o This figure indicate that the RSF can vary by ~1 order of magnitude over the course of the day. While the difference between gaining and loosing conditions is and interesting result. How do you justify this with the range of stream orders, sediment size, and hydraulic conductivity show in the Gomez-Veles papers?
  - o Are you using the stream order of the USGS gauge you gather the data from? If so report these information and explain this process in the text

  **Response:** Thank you for the questions. As we responded to the second specific question, the RSF values were calculated with equation 8, where stream orders, sediment size and hydraulic conductivity were not variables determining the values. However, variables in equation 8, such as the discharge Q, are directly influenced by the geomorphological settings. Studying these influences is beyond the research scope of the present paper.

- Discussion
- Line 267, Water table drawdowns coupled with hydraulic gradient changes through temperature contribute to enhanced diel fluctuations of exfiltrating hyporheic fluxes
  **Response:** Thank you for the suggestion. We modified the sentence as suggested.

- Line 269, Under the neutral condition
  **Response:** Modified as suggested.

- Line 272, 269 –
  - You only reference figure 3 here which is the gaining condition, should you also mention figure 5 the loosing condition?
  - Or be more specific in the text

    **Response:** In line 269, neutral conditions were only plotted in figure 3 and not in figure 5. In line 272, both gaining and losing conditions were referenced.

- Paragraph on Line 285
  - I agree with what you are saying
  - Don't pump an aquifer during a storm because the drawdown could pull pollutants into the hyporheic zone
  - Could you provide an example of a usgs site that has daily drawdowns from groundwater pumping like the ones shown in this paper from the plants?
  - This may be a hard reach but could have important management implications

    **Response:** Thank you for the question. Daily groundwater table fluctuations were conceptualized as sinusoidal curves with varying amplitudes and phases, which were not observations in USGS sites. Limitations of this simplification were discussed in section 4.5 from line 378 to 393. In the present study, only river discharge and temperature time series are observations in USGS gauging stations.

- Line 307, could you use your data (from figure 6 maybe) to show this?
  - Upwelling keeps warm surface water from connecting to HZ

    **Response:** Thank you for this question. As we responded to the reviewer's comment on Figure 6, a gif figure will be added to illustrate the dynamics of temperature fields in the sediment. In this figure, hyporheic zones can be completely compressed by upwelling fluxes when the river stage is low, which prevents the warm surface water from penetrating into the sediment especially during summer.

- Therefore, in summer when river temperature is relatively high, the hydraulic conductivity is enhanced and becomes the main modulator for hyporheic exchange rate under losing condition.
    - Change the therefore language. The authors use this word a lot

        **Response:** Thank you for the suggestion. "Therefore" is replaced by "consequently".

- Combine the paragraphs between Lines 310 and 320

    **Response:** Changed as suggested.

- Line 343, Therefore, hyporheic zones have a larger cooling effect during high river temperature under out-of-phase gaining conditions than under in-phase conditions (under gaining conditions)

    **Response:** Added as suggested.

- Too many conditions maybe think of different wording for in-phase and out-of-phase (conditions)

    **Response:** We replaced a couple of "conditions" with "scenarios".

- Loosing conditions speeds up residence time (RSF = reaction scale factor)

    **Response:** Reaction significance factor is proportional to the residence time (equation 8). $RSF_a$ under gaining conditions is around three orders of magnitude higher than under losing conditions due to the significantly longer residence time resulting from mixing between surface water and groundwater under gaining conditions.

- Gaining conditions slows down residence time and allows mixing of GW and SW

    **Response:** Groundwater has significantly longer residence time. The mixing between groundwater and surface water under gaining conditions thus increases

the mean residence time of the exfiltrating hyporheic fluxes. Therefore, $RSF_a$ under gaining conditions is higher than under losing conditions.

- In conclusion, the timing 365 of groundwater table drawdown is more important under gaining conditions than under losing conditions for denitrification reactions.

  **Response:** Yes. With groundwater losing conditions, even though $RSF_a$ display peaks on a logarithmic scale, the actual differences of $RSF_a$ (in the scale of 10 to the power of −5) between in-phase and out-of-phase conditions are insignificant compared to gaining conditions.

- Line 668 – could you mention this fact earlier in the paper, so the reader is not thinking about denitrification the entire time?

  **Response:** Thank you for the suggestion. The scaling of RSF for different reactions will be explained as soon as the RSF is first introduced in line 260:

  *It's worth noticing that instead of the denitrification, reaction potential of a different geochemical process can be assessed if a different characteristic time scale is applied in equation 8.*

- Study limitations?
  - What about connectivity? A reference to some of this great work would be nice to see in this paper

    **Response:** Thank you for the suggestion. We added a short paragraph discussing hydrological connectivity at the end of discussion. Please refer to the specific comment #3.

- Conclusion
  - Not strong enough or long enough
  - Need more space and references to specific aquatic community impacts and groundwater table diel drawdown.

**Response:** Thank you for the comment. The conclusion is rephrased as below:

*Groundwater table dynamics substantially modulate hyporheic exchange processes. Daily groundwater table drawdown causes additional variability of hyporheic exchange besides the variability induced by the diel river temperature fluctuations. However, the variability induced by daily groundwater table drawdown is not necessarily an addition to the fluctuations induced by the diel river temperature changes. More specifically, groundwater flow fluctuations that are out-of-phase to diel river temperature fluctuations are likely to promote hyporheic exchange to a larger extent than groundwater flow fluctuations that are in-phase to diel river temperature fluctuations. Even though both groundwater table fluctuations and diel river temperature fluctuations affect hyporheic exchange dynamics, under the same discharge condition river temperature has a more dominant role in determining hyporheic exchange variability under losing conditions than under gaining conditions. This is because under gaining conditions, heat advection of upwelling groundwater is more dominant; under losing conditions heat advection and conduction of surface water is more dominant in hyporheic zone's heat exchange.*

*The timing of groundwater table drawdown modifies the rates of hyporheic exchange, and as a result the mixing and spreading of pollutants in the aquifer. The timing of aquifer pumping should be adjusted to avoid flood events in order to ensure minimal contaminant uptake. Additionally, the timing of groundwater table drawdown also affects the hyporheic zone's ability to act as a temperature buffer that protects aquatic communities from thermal extremes. Although not as significant as the effect of flood events, hyporheic denitrification potential (and potentially for other biogeochemical reactions) is also changing following the groundwater table drawdown. Therefore, careful considerations must be taken when planning aquifer pumping schedules in order to minimize negative environmental impacts.*

---

## Referee Comment (RC2) · Anonymous Referee #2 · 9 Oct 2020

The authors present an extensive modelling study on the interplay between diurnal temperature effects and groundwater gradients on the dynamic evolution of the hyporheic zone in a river with a defined bedform topography. The hyporheic zone is a highly relevant transition zone controlling biogeochemical processes such as denitrification in streams (e.g., Gomez et al. (2015)). Therefore, the topic of the manuscript fits well with the scope of HESS.

The processes affecting the exchange between river water, the hyporheic zone and groundwater are highly non-linear and can lead to seemingly counter-intuitive effects. The authors build on previous work (e.g., Wu et al., (2020, 2018)) and a model to investigate the questions specific to this manuscript. In particular, they study how daily temperature fluctuations in a stream impact the hyporheic exchange and how it interfers

with effects caused by dial fluctuations of groundwater fluxes caused by evapotranspiration or pumping.

The authors provide a broad range of data and results on the hyporheic water fluxes, temperature gradients and potential impacts on biochemical process rates such as denitrification.

The manuscript is interesting. But before it can be published I suggest major revisions for clarifying open issues and for improving the structure to enhance readability.

**Major issues**:

**Improve readability** The structure of the text is not always very reader-friendly. This means that it is not always easy to immediately understand and follow the logic of the arguments and results. This observation holds true for single paragraphs as well as for entire sections (e.g., the Result section). Often the starting point of an argument is not what is directly evident to the non-specialists but the necessary explanations follow only afterwards.

The text on L. 55 - 58 may serve as an illustrative example: The starting point is that there are diel fluctuations of hyporheic exchange and that they may interact with diurnal changes of groundwater fluxes. However, for the non-specialist regarding the hyporheic zone, the diel fluctions may not be evident. Hence, upon reading one stops and reflects why this should be the case. In the current manuscript, the explanation comes only afterwards. I suggest a different structure:

1. Daily temperature fluctuations in stream (every reader will know and agree)
2. This affects viscosity and hence hydraulic conductivity (the readers will follow)

[Figure]

3. This induces diurnal changes in hyporheic exchange as demonstrated in Wu et al. (2020) (the reader will believe this)

4. There are also diel fluctuations in groundwater fluxes for several reasons (readers will know and agree)

5. Therefore, there are two dynamic processes affecting the hyporheic zone and they may potentially interact in rather non-linear ways.

This is just an example but I suggest to pay due attention to this aspect because the authors claim (with good reasons) that hyporheic processes have wider implications. This means their paper should also be read by a wider audience in the hydrology and water resources management community. Accordingly, they should write the paper for such an audience and consider what to expect from such readers as starting points for presenting the arguments and results.

**Model description** There are several aspects of the model and its set-up that are not fully satisfactory:

1. *Model dimensions.* Given that the authors have used a 2-D model (L. 81), the model domain has to have dimensions along the x- and z-axes. Please provide this information (e.g., in terms of $\lambda$). Please demonstrate as well that this model set-up is a meaningful representation for the case study that represents a given real situation.

2. *Fig. 2.* At that point, the panels *b* and *c* are rather confusing. Panel *a* is very generic, but on the lower panels real dates are given and it is not clear to the reader what these values on the x-axes mean and why the are chosen. It is also obscure what the temperature represents. It takes a lot of reading until one can make the link to the case study and the respective observations.

3. *Mass balance.* From Fig. 2 (a), it follows that the water balance for the model domain is given by $Q_{river-out}(t) = Q_{river-in}(t) + q_b(t)$. Based on

how the boundary conditions are defined however, the water flow in the river is independent on the groundwater fluxes imposed (the flow simply follows from the prescribed $H_s(t)$ (Eq. 2, 3). Also the head distribution at the water-sediment interface is flux-independent. However, this distribution was derived from empirical observations Elliott & Brooks (1997) without considering gaining or losing situations. This seems to be adequate as long as $U_s(t) H_s(t) >> q_b(t) L_{domain}$ with $L_{domain}$ being the length of model domain. Please i) provide the evidence that this holds true for the case study and the dimension of the model domain, and ii) make these aspect also clear in the discussion. Actually, this aspect seems to emphasis the importance of the findings: even small groundwater fluxes may have a pronounced influence on the hyporheic zone. This may be evident to the authors, but I missed that point in the context of the entire paper.

4. *Eq. 6a*. I could not find an explanation for $a_0$. It is tedious to go to previous publications and guess that $a_0 = 1$.

5. *Model implementation*. Please provide some information on the model implementation (grid set-up, model version, run time etc.).

6. *Defining the hyporheic zone*. It is unclear how the procedure described on L. 130 - 136 is actually implemented. First, because the hyporheic zone changes over time, the proposed procedure needs to be repeated, I assume. Can you comment on that? Second, for neutral and losing conditions, it seems that the threshold $C \geq 0.9 C_s$ will eventually exceeded across the entire domain. Can you clarify?

**Description of the case study** This description is very superficial and has to be improved substantially.

1. *Site identification and description* Please provide more information on the site including the location and name. It is not necessary that every interested reader has to check the USGS website. Describe some key characteristics of the climate and hydrology of the catchment and the measuring site (altitude, mean discharge etc.). This is important to put the findings in a proper context.

It is also essential to know which observation period was used for the simulations. One learns only at a later stage (e.g., from Fig. 3a) that three hydrological years seem to have been used.

On L. 160, the amplitude of groundwater flux changes are linked to a range of the groundwater table fluctuations. Although a reference is provided, this is not sufficient. Boano et al. (2008) presents a general framework for linking stream-groundwater interactions and the influence on the hyporheic zone, but not any site-specific information for this case study. Describe the approach including the equations used and the model assumptions. In this context, it would be also useful to provide evidence that this assumed water table fluctuation is also reasonable for a hypothetical groundwater pumping operation.

The paragraph on L. 144 - 155 describes the *in-phase* and *out-of-phase* conditions. It might enhance the intuitive understanding for a general reader if the authors indicate more explicitly that the *out-of-phase* conditions represent the natural state with high stream temperatures and lower water table in the aquifer due to transpiration by the vegetation.

**Result section:** This section contains a lot of material (which is positive) but the way of presenting needs improvement. The more so because not all of the necessary results seem to be shown so far.

1. *Structure* One of the key messages of the manuscript is that there is an intricate interplay between the temperature regime, the flow regime of the stream and the water table fluctuations in the aquifer that needs to be understood. To be able to understand this, one has to get an overview about the general conditions prevaling at the study site during the period of interest. Therefore, I suggest to start with a short description of the key features of the three hydrological years.

Subsequently, it helps the reader if the complexity is increased in a stepwise fashion. Therefore, I would first describe the results for the neutral conditions, then the losing conditions and finally the gaining conditions. Furthermore, I suggest to use explanations such as on L. 277 - 279 to frame the result section in a way that is intuitive also to the non-specialist reader.

2. *Nomenclature* One of the confusing things is the terminology used for describing the hyporheic fluxes. Nowhere it is explained what actually meant by the infiltrating and exfiltrating hyporheic fluxes. For the neutral case, the two fluxes are identical, which makes sense. Under gaining conditions, the infiltrating flux is consistently larger than the exfiltrating flux. How is this explained and why is the same true for the losing conditions when there is a net flux from the river to the aquifer? Please clearly define the terms and explain the apparent contradictions mentioned.

3. *Residence times* The method sections describe how to estimate time-variable residence times in the hyporheic zone. Despite of using an average value for calculating the reaction significance factor RSF, no data on residence times are provided. This is essential if one would like to be able to evaluate the relevance of the results for any biological or bio(geo)chemical processes. Provide the results on the time-variant residence times and how they change upon the different boundary conditions.

4. *RSF* First of all, this approach has not been introduced so far. It should be mentioned in the Introduction when introducing the denitrification topic and described in the method section. Apart from that I am not sure whether the chosen form is an adequate implementation of the concept. I have three

question marks:

(a) The first relates to $q_{HZ}$ because I could not follow what this term actually represents (see above: how does it relate to infiltrating and exfiltrating fluxes?).

(b) Why is the mean residence time used for calculating a time-variant quantity such as RSF when residence times were derived as a function of time? Depending on the temporal correlation functions between the relevant hyporheic flux $q_{HZ}$ and the residence times $\tau_{HZ}$, there might be substantial deviations from the current version.

(c) The time scales of denitrification. First, the description of how $\tau_{HZ}$ was parameterised is insufficient. Which quantiles in Gomez et al. (2015) do you refer to? Second, denitrification depends very much on temperature (e.g., Boulêtreau et al. (2012)). This implies that $\tau_{dn}$ is not constant. Given that the manuscript deals with temperature as a key influencing factor, it would seem logic to consider such a temperature dependence also for $\tau_{dn}$. At least one could test the sensitivity of RSF against the temperature dependence of denitrification.

5. *Plausibility check against empirical data* One of the values of such a model study is the possibility to study processes and their interactions under well defined conditions and to explore system behaviours that are otherwise impossible to obtain. This comes at the costs of the difficulty to relate the model findings and insights back to the real world. To improve on that the authors should provide more context on the case study (see above). On the other hand, they should also add some comparisons of model results with empirical observations to provide some plausibility checks. Possibilities for doing so would for example be the extent of the hyporheic zone, residence times (both not even shown for the model results, see above) or RSF values as depicted in Fig. 8. Such values could for example be compared to estimates provided by Gomez et al. (2015).

**Detailed comments:**

**L. 18 - 19:** Why is this understanding *key to water resources management*? There are many aspects relevant for water management (land use management, hydropower generation schemes etc.). Please be more specific for aspects this understanding is key and why.

**L. 23, 26 and elsewhere:** Articles or pronouns are missing sometimes. Please have a linguistic check.s

**Fig. 4:** Explain the time axes and give a reason why only that part of the entire study period is displayed? It seems to be rather arbitrary. Are the results from the *in-phase* or *out-of-phase* simulations?

**Fig. 6:** Unfortunately, one can hardly see the differences between *a* and *b* or *c* and *d*, respectively. One option could be to show the respective difference plots and to add difference plots for the fluxes.

**Fig. 8:** Add the year to the time axes and explain why this specific period was selected.

**References**

Boano, F., R. Revelli, and L. Ridolfi. 2008. Reduction of the hyporheic zone volume due to the stream-aquifer interaction. Geophysical Research Letters 35.

Boulêtreau, S., E. Salvo, E. Lyautey, S. Mastrorillo, and F. Garabetian. 2012. Temperature dependence of denitrification in phototrophic river biofilms. Science of the Total Environment 416:323-328.

Elliott, A. H., and N. H. Brooks. 1997. Transfer of nonsorbing solutes to a streambed with bed forms: Theory. Water Resources Research 33:123-136.

Gomez-Velez, J. D., J. W. Harvey, M. B. Cardenas, and B. Kiel. 2015. Denitrification in the Mississippi River network controlled by flow through river bedforms. Nature Geoscience 8:941-945.s

Wu, L., J. D. Gomez-Velez, S. Krause, T. Singh, A. Wörman, and J. Lewandowski. 2020. Impact of Flow Alteration and Temperature Variability on Hyporheic Exchange. Water Resources Research 56:e2019WR026225.

Wu, L., T. Singh, J. Gomez-Velez, G. Nützmann, A. Wörman, S. Krause, and J. Lewandowski. 2018. Impact of Dynamically Changing Discharge on Hyporheic Exchange Processes Under Gaining and Losing Groundwater Conditions. Water Resources Research 54:10,076-010,093.

---

## Author Response (AR1)

Dear Dr. Christian Stamm,

Thank you for your positive feedback on our manuscript entitled "How does daily groundwater table drawdown affect the diel rhythm of hyporheic exchange?" [MS No.: hess-2020-288]. The comments from the reviewers are very insightful. We have addressed all the points raised by both reviewers. To summarize, the following major changes are made:

- Readability is improved. Modifications are made mainly in the Introduction, Results, and Discussion sections.

- Key definitions are clarified. These include hyporheic zone boundary, mean residence time, reaction significance factors, infiltrating and exfiltrating hyporheic fluxes.

- Supplementary information is added. A section describing the calculation of groundwater table drawdown, and two figures showing mean residence time and characteristic time scale for denitrification are included.

We look forward to your response.

Kind regards,

Liwen Wu

c.c.: Jesus D. Gomez-Velez, Stefan Krause, Anders Wörman, Tanu Singh, Gunnar Nützmann, and Jörg Lewandowski

**Response to Comments from Referee #1**

**Initial Comments**

The authors of this paper use USGS gauge data with diel fluctuations in discharge and river temperature to model hyporheic exchange rates in order to better understand how daily groundwater table fluctuations change hyporheic exchange rates in gaining, losing, or neutral streams. The authors use complex modeling to show how in-phase or out-of-phase daily groundwater table drawdown can influence hyporheic exchange rates. The model created for this paper makes hard assumptions about river morphology, network position, and sediment characteristics to step back and look at daily groundwater table dynamics conceptually. While much of the paper is modeling hyporheic exchange the authors also ask how diel groundwater table fluctuations and river temperature impact residence time for denitrification potential and thermal refugia for aquatic species. The authors conclude that groundwater table dynamics modulate hyporheic exchange process differently than diel river temperature. When diel groundwater table drawdown is out-of-phase with river temperature hyporheic exchange is greater than when in-phase. Under gaining conditions upwelling groundwater buffers diel river temperature and increases hyporheic exchange rates. Under losing conditions surface water temperature penetrates deeper into the hyporheic zone and decreases hyporheic exchange rates. The authors do a good job in the modeling and data analysis sections of this research yet need to make the objectives of this paper clearer to support the data presented in this paper.

> **Response:** Thank you for the positive comment on the modeling and data analysis, and also for the insightful suggestions on improving the manuscript. The point-by-point reply to the comments is given below. Changes in the manuscript are indicated by _underlined text in italic_. Line numbers in this response refer to the numbers in the track-changed manuscript. By responding to the following comments, we incorporated the changes to clarify the objectives, model assumptions, and to improve the structure for a better readability.

**Specific Comments**

**R1_1** The objective statement of this paper is not well defined. After a good introduction, the last paragraph is lacking in clarity as to what this paper is about. Suggestion for the authors to use language like: "In the present study, we aim to quantify the impact of groundwater withdrawal on hyporheic exchange processes at the daily scale as well as better understand impacts on potential denitrification and thermal buffering". Then move on to how this paper accomplished the objectives. "To investigate these objectives we built a complex model that…." This will also help guide the reader towards the start of the methods section.

> **Response:** Thank you for the good suggestion. To better present the objectives, we modified the last paragraph of the introduction (from Line 65-69) in the following way:
>
> _"In the present study, we aim to quantify the impact of river temperature fluctuations and groundwater table drawdown on hyporheic exchange processes at daily scales, as well as to better understand implications on hyporheic zone's potential for denitrification and thermal buffering. With these objectives in mind, different groundwater scenarios corresponding to different timings of groundwater table drawdown under gaining and losing conditions are applied in a physically based hyporheic flow and heat transport model."_

**R1_2** The connection from the modeling to RSF and thermal refugia for aquatic species is weak. It feels like the nutrient processing and ecosystem services provided by hyporheic exchange are tossed into this paper to try to broaden the scope of the paper. I suggest that the authors leave nutrient processing to the discussion section rather than a main objective of this paper. Much of the paper does good modeling of hyporheic exchange rates and that should be the focus. There is also some confusion in if this paper wants to just focus on denitrification or RSF and this distinction needs to be clear to the audience. The authors also provide no hard numbers as to how RSF was applied to their model. The Gomes-Velez (2016) paper provides a range of RSF for stream orders 1-12 and how RSF varies throughout stream orders. The authors fail to mention what RSF values were chosen

amongst that range. While the result of the RSF analysis is interesting, the explanation as to what this mean ecologically is missing.

**Response:** To better address this comment, we will answer the three sub-comments in the following order:

1) How was RSF calculated? Were RSF values chosen from Gomez-Velez (2016)?

The RSF values were not chosen from Gomez-Velez (2016). They were calculated under the specific flow and sediment characteristics in the present study by Eq 8 which was first introduced by Harvey et al. (2013):

$$RSF_a = \frac{q_{HZ}}{Q} \cdot \frac{\tau_{HZ}}{\tau_{dn}} \tag{Eq. 8}$$

where $Q$ is the river discharge, $q_{HZ}$ is the exfiltrating hyporheic flux calculated with Eq 1-3, $\tau_{HZ}$ is the mean residence time of hyporheic flow calculated with Eq 6, $\tau_{dn}$ is the characteristic time scale for denitrification determined based on Gomez-Velez and Harvey (2014) and Gomez-Velez et al. (2015).

To better present the values of $\tau_{dn}$, we will add the following figure in the supplementary information to show the quantiles of the characteristic time scales for denitrification.

[Figure]

*Figure S1: Box plot of the characteristic time scale for denitrification ($log_{10}[h]$). The 25th quantile is 0.38, the 50th quantile is 0.87, and the 75th quantile is 1.28. (Taken from Gomez-Velez and Harvey (2014) and Gomez-Velez et al. (2015))*

In the manuscript, the following text was added in Line 281:
*The 25th, 50th, and 75th quantiles are presented in Fig. S1 in the supplementary information.*

2) Does the paper focus on denitrification or RSF? What are the differences?

We calculated the reaction significance factors for denitrification with equation 8. However, the interpretations of results shown in Figure 8 are not limited to denitrification processes. For a different biogeochemical reaction, another characteristic time scale is applied instead of $\tau_{dn}$. Results presented in Figure 8 will only be scaled by a different biogeochemical time scale for the reaction of interest. The relative variations of RSF remain the same for other biogeochemical reactions (as stated in line 401-404).

3) Are nutrient processing and ecosystem services main focuses of this paper?

The ecosystem service is not the main focus of this paper. The impact on ecosystem is not quantified but only discussed in the Discussion section to show the impact of timing of groundwater table drawdown on the hyporheic zone's function as thermal buffers for aquatic communities qualitatively. However, the nutrient processing as we presented for denitrification was quantified with equation 8 and the results are presented in Figure 8. We think the results have clearly demonstrated the different impacts of groundwater table fluctuation on reaction potentials under gaining and losing conditions, which is a worthwhile message for readers who are interested in exploring biogeochemical reactions under different

groundwater conditions. Therefore, we would like to keep the quantification of the reaction potential in the main objective.

**R1_3** Hyporheic connectivity is not discussed or mentioned in this paper. How does connectivity change during these diel fluctuations or during storms? How connected the hyporheic zone is could impact the thermal buffering capacity. A short paragraph on this topic should be added.

> **Response:** This is a good point. We added a short paragraph about hyporheic connectivity at the end of discussion in line 409:

> *The first term in $RSF_a$ ($\frac{q_{HZ}}{Q}$) describing the proportion of the river discharge passing through the hyporheic zone per unit bedform area can be used to quantify the connectivity between river and hyporheic zone (Harvey et al., 2019). This connectivity underpins many ecosystem processes and important reactions that take place in close contact with biogeochemical reactive sediments (Boulton, 2007; Ward et al., 2000; Malard et al., 2002; Roley et al., 2012). Maintaining a good hydrological connectivity is therefore crucial. Under the same river discharge rates (Q), hyporheic exchange rates ($q_{HZ}$) are higher when groundwater drawdown is in an out-of-phase pace with diel river temperature fluctuations than in an in-phase pace. Consequently, the hydrological connectivity is higher in a groundwater out-of-phase scenario. The temperature differences between river and exfiltrating hyporheic fluxes with in-phase and out-of-phase groundwater table drawdown also proves this finding (Fig. 7). Hydrological connectivity is higher in out-of-phase groundwater table fluctuation scenarios than in in-phase scenarios, making hyporheic zone a better thermal buffer under out-of-phase scenarios.*

**R1_4** The is also confusion as to what a groundwater table drawdown means. The diel groundwater fluctuations presented here are due to plant uptake, yet the authors also mention groundwater pumping. The introduction paragraph (Lines 64-70) sets up the

pumping problem well but does not mention plants. The discussion section does not discuss the pumping problem well enough to support the management implications in the conclusion. The implications for poorly designed pumping schedules are huge given your data during the flood event!

**Response:** Thank you for the comments. As introduced in line 46-48, both the phreatophytes water-use and anthropogenic pumping can cause groundwater table drawdown at a daily scale. Therefore, the daily groundwater table drawdowns in the present study were conceptualized as sinusoidal curves with varying amplitudes and phases.

We do understand the reviewer's confusion on line 64-70. To clarify the research set up, the description of pumping problem will be removed from introduction. This issue will instead only be discussed in Discussion to present the implications of our results on pumping management. A short paragraph will be added in line 311:

*Modern regulating reservoirs are usually designed with enough storage capacities allowing planning of pumping schedules independent of user demand (Reca et al., 2014). A poorly designed pumping regime is detrimental to the biological and ecological functioning of the fluvial systems (Moore, 1999; Libera et al., 2017; Bredehoeft and Kendy, 2008). Consequently, careful selection of aquifer pumping schedules with considerations of both timing of flood and groundwater table dynamics are critical for water management agencies to minimize the environmental footprint of the withdrawal process.*

**R1_5** The conclusion is also weak and doesn't drive home the answers found from the objective statement. The closing sentence is subjective and needs to be reworded: "Our data show that hyporheic exchange rates in a gaining river increase significantly during storm events. When combined with an in-phase diel groundwater table fluctuation, hyporheic exchange rates are higher than an out-of-phase fluctuation (Fig 3f storm vs. Fig 5f storm). Anthropogenic aquifer pumping schedules should be out of phase with diel river

temperature to ensure minimal contaminant uptake". RSF or denitrification also needs to me worded stronger here.

> **Response:** Thank you for the comment. The conclusion has been rephrased. Please find it in the response to the last comment on the Conclusion (**R1_47**).

**R1_6** Transitional sentences between paragraphs and sections need to be stronger making it hard for the reader to follow

> **Response:** Thank you for the comment. We rephrased the last paragraph of the Introduction and the Conclusions to improve the connections between sections. Please refer to responses to comments on **R1_1** and **R1_47**.

**Technical Comments**

**R1_7** Abstract ok

- The phrasing of groundwater withdrawal makes it sound like there is anthropogenic influence. You do not specifically look at this so I would keep it to the discussion section

> **Response:** Done as suggested. Groundwater withdrawal is replaced by groundwater level drawdown to reflect the groundwater level fluctuation in a more general way. The text in line 10 now reads:
>
> *The timing of groundwater table drawdown has a direct influence on hyporheic exchange rates and hyporheic buffering capacity on thermal disturbances.*

**R1_8** Line 14, I would turn this first sentence in a strict definition of the hyporheic zones

- Something like hyporheic zones are transitional areas between surface water and groundwater environments that often exhibit marked physical, chemical, and biological gradients that drive the exchanges of water flow, energy, solute and microorganisms between surface and subsurface regions.

o   This will help focus the readers the research in this paper

> **Response:** Thank you for the suggestion. We modified the sentence as suggested in line 14:
>
> *Hyporheic zones are transitional areas between surface water and groundwater environments, which often exhibit marked physical, chemical, and biological gradients that drive the exchanges of water flow, energy, solute and microorganisms between surface and subsurface regions.*

**R1_9** Line 18, what makes researching spatiotemporal variability of hyporheic exchange key to water resources management? Provide a reference

> **Response:** Thank you for the question. We modified this sentence to have more connections with the previous sentence:
>
> *Understanding the spatiotemporal variability of hyporheic exchange processes is key to characterizing the nutrient cycling and river ecosystem functioning (Lewandowski et al., 2019)*

**R1_10** Line 19, what and how is it key to ecosystem restoration

> **Response:** Please refer to **R1_9**.

**R1_11** Line 23, change to factors influencing the hydraulic….

> **Response:** Done as suggested.

**R1_12** Line 26, change language. Make this more clear

> **Response:** Please refer to the next comment (**R1_13**) below.

**R1_13** Entire second paragraph needs to be worded better

**Response:** Entire second paragraph is re-worded as below:

*Hydrological drivers and modulators of time-varying hyporheic exchange processes have been extensively studied in the last decade. The hydraulic gradient as the main driver of hyporheic exchange processes is changing along the sediment-water interface, determining (1) the spatiotemporal variability of hyporheic zone extents and (2) characteristic time scales of hyporheic exchange (Boano et al., 2013; Ward et al., 2017; Gomez-Velez et al., 2017). Factors influencing the hydraulic gradient at the sediment-water interface include channel flow (Trauth and Fleckenstein, 2017; Grant et al., 2018; Broecker et al., 2018), geomorphological settings (Tonina and Buffington, 2011; Schmadel et al., 2016; Singh et al., 2019), and regional groundwater flow (Nützmann et al., 2014; Malzone et al., 2016; Wu et al., 2018). Sediment and fluid properties do not drive hyporheic exchange, but they modulate hyporheic exchange substantially: sediment heterogeneity can alter hyporheic flow paths and residence time distributions, creating hot spots for biogeochemical transformations (Sawyer and Cardenas, 2009; Gomez-Velez et al., 2014; Pescimoro et al., 2019); fluid properties, i.e., density and viscosity, are functions of temperature and directly influence the hydraulic conductivity, thus hyporheic flow. Consequently, river temperature variability (i.e., diel and seasonal river temperature fluctuations) induces significant changes of hyporheic exchange processes (Cardenas and Wilson, 2007a). The spatiotemporal variability of the drivers and modulators eventually results in dynamic hyporheic exchange processes. Among these drivers and modulators, the combined effects of regional groundwater flow and river temperature on dynamic hyporheic exchanges are comparably understudied.*

**R1_14** Line 43 – Good sentence here

**Response:** Thank you!

**R1_15** Figure 1

- Groundwater table A and B separation is confusing to the eye

  **Response:** To better separate groundwater table A and B, we colored differently for groundwater table A and B.

- Do these relate to either the gaining or losing condition

  **Response:** Yes, groundwater table A refers to gaining condition where the groundwater table is higher than river stage; groundwater table B refers to losing condition where groundwater table is lower than river stage.

- Suggestion to color the lines differently

  **Response:** Done as suggested.

- Remove the tree image or add more. Suggestion to use a tree silhouette.

  **Response:** The tree was removed.

**R1_16** Line 45, reference needed for 1st sentence

  **Response:** The sentence in line 45 shares the same references with the following sentence. The references are hence stated at the end of the next sentence in line 48.

**R1_17** Line 58, Wu et al. observed….

  **Response:** Done as suggested.

**R1_18** Line 71, This entire paragraph needs to be stronger

  **Response:** This paragraph is rephrased. Please refer to **R1_1** and **R1_19**.

**R1_19** Transition from objective statements to modeling section is poor

- o Ideas for objective statements
  - ▪ Stronger, need to be more focused. This paragraph is short and weak when it should be the strongest hit of the paper
    - • In the present study, we aim to quantify the impact of groundwater withdrawal on hyporheic exchange processes at the daily scale as well as better understanding river temperature impacts on potential denitrification and thermal buffering.

  **Response:** Thank you for the suggestion. We have modified this as suggested in line 65.

- o Modeling transition
  - ▪ Use the last paragraph to transition to the modeling
    - • This is poor

  **Response:** Thank you for the suggestion. The following sentences are added to act as a transition to the modeling section from line 67:

  *With these objectives in mind, different groundwater scenarios corresponding to different timings of groundwater table drawdown under gaining and losing conditions are applied in a physically based hyporheic flow and heat transport model. Hyporheic exchange rates, temperature distribution and denitrification efficiency are quantified to assess the impacts of river temperature and groundwater level fluctuations on hyporheic exchange processes.*

**R1_20** Line 80, need a transition sentence to connect to the aims

**Response:** The following sentence is modified in line 76 to connect to the aims:

*To understand the hyporheic exchange in response to changing river discharge, temperature and groundwater table fluctuations, a two-dimensional conceptualization is proposed based on Wu et al. (2018) and Wu et al. (2020) (Fig. 2a).*

**R1_21** Line 84, need reference for COMSOL method and mesh-independent.

**Response:** The COMSOL model was developed based on Wu et al. (2018) and Wu et al. (2020) (Fig. 2a) as stated in line 77.

**R1_22** Figure 2

o Good conceptual figure

**Response:** Thank you!

**R1_23** Figure 3

o Say that discharge is not to scale, rather than not labeled. Or that you are using it for visual aid and not to scale

**Response:** Thank you for the suggestion. The figure caption is modified as below:

*For figure clarity, discharge is not scaled in e and f, and used only for qualitative comparisons.*

**R1_24** Line 214, you say only in-phase results are shown but Figures 3 and 5 show out of phase results

**Response:** Thank you for the comment. Effects of groundwater table fluctuation amplitudes on dynamic hyporheic responses are only explored under in-phase scenarios, because under out-of-phase scenarios fluctuations of exfiltrating hyporheic fluxes are almost always in the same phase with the diel river

temperature fluctuations. Therefore, unlike in-phase scenarios, the phase shifts due to reduced amplitudes in groundwater table fluctuation are not observed. Reduced amplitudes in groundwater table fluctuation under out-of-phase scenarios only contribute to reduced amplitudes in exfiltrating hyporheic flux fluctuations. Based on these reasons which are also stated in line 230-235, only results in in-phase scenarios are presented in Figure 4.

To clarify in the text, the following sentence is modified in line 235 as below:

*For simplicity, only results in in-phase scenarios are presented in Fig. 4.*

**R1_25** Figure 4

o   I don't like the positioning of Figure 4 but don't know if you have control over this or the journal does. It looks odd to have a figure showing gaining conditions in the 3.1.2 under Losing Conditions section of the paper

**Response:** Thank you for the suggestion. Unfortunately we are afraid that positions of figures are beyond our control in the final published version.

**R1_26** Figure 5

o   Caption says discharge is not labeled when it is in Fig 5c and Fig 5d

o   I think you may mean that discharge is not to scale in 5e and 5f

**Response:** The caption is correct in referring to Fig 5c and Fig 5d. There are no Fig 5e and 5f.

**R1_27** Line 260, please state the values you used for you models or at least a range of values

**Response:** These values are stated in response to the second specific comment (**R1_2**) and presented in Figure S1.

**R1_28** Figure 6

- I'm not sure how necessary figure 6 is in this paper. While I like the figure, I believe you could and do explain this information in the text.
- This could help you shorten the paper
- You could slow spice this up by clipping a few of these snapshots together and then playing them in a .gif over the course of a storm so you could see the variations in the losing condition sections of the figure

**Response:** Thank you for this comment. We modified Figure 6 by only keeping 2 sub-figures in order to illustrate the differences between gaining and losing conditions. The differences between in-phase and out-of-phase scenarios can be explained using Fig. 7. In this way, the paper can be shortened without losing important information.

**R1_29** Figure 7

- Same weird out of place figure placement
- I like this figure. It tells me clearly that gaining in-phase hyporheic zones have less variable temperature from the constant upwelling of groundwater
- Get rid of the underscore in gaining in-phase, keep it consistent with the figures above. Same goes for the color scheme if possible

**Response:** Thank you for the suggestion. We have done as suggested. The color scheme has not been changed because the color contrast is not obvious if using the same color scheme as previous figures under the current plot.

**R1_30** Line 260

- Gomez-velez et al 2015 reports RSF over entire river networks. How you are you implementing these findings into this new model? The also include river bedform information and this paper assumes uniform sediment. So please list what metrics you are using from this Gomez-velez paper. What are the quantiles???

**Response:** We have addressed this comment in **R1_2**.

**R1_31** Figure 8

- o Under loosing conditions reaction significance time is 3 orders of magnitude less than gaining conditions

- o This figure indicate that the RSF can vary by ~1 order of magnitude over the course of the day. While the difference between gaining and loosing conditions is and interesting result. How do you justify this with the range of stream orders, sediment size, and hydraulic conductivity show in the Gomez-Veles papers?

- o Are you using the stream order of the USGS gauge you gather the data from? If so report these information and explain this process in the text

  **Response:** Thank you for the questions. As we responded to the second specific question (**R1_2**), the RSF values were calculated with equation 8, where stream orders, sediment size and hydraulic conductivity were not variables determining the values. However, variables in equation 8, such as the discharge Q, are directly influenced by the geomorphological settings. Studying these influences is beyond the research scope of the present paper.

Discussion

**R1_32** Line 267, Water table drawdowns coupled with hydraulic gradient changes through temperature contribute to enhanced diel fluctuations of exfiltrating hyporheic fluxes

    **Response:** Yes, that's a good summary.

**R1_33** Line 269, Under the neutral condition

    **Response:** Modified as suggested (line: 285).

**R1_34** Line 272, 269 –

- o You only reference figure 3 here which is the gaining condition, should you also mention figure 5 the loosing condition?

- o Or be more specific in the text

  **Response:** In line 286, neutral conditions were only plotted in Figure 3 and not in Figure 5.

**R1_35** Paragraph on Line 285

- o I agree with what you are saying

- o Don't pump an aquifer during a storm because the drawdown could pull pollutants into the hyporheic zone

- o Could you provide an example of a usgs site that has daily drawdowns from groundwater pumping like the ones shown in this paper from the plants?

- o This may be a hard reach but could have important management implications

    **Response:** Thank you for the suggestion. Daily groundwater table fluctuations were conceptualized as sinusoidal curves with varying amplitudes and phases, which were not observations in USGS sites. Researches on pumping induced groundwater drawdown can be found in Butler et al. (2001 & 2007), etc.

**R1_36** Line 307, could you use your data (from figure 6 maybe) to show this?

- o Upwelling keeps warm surface water from connecting to HZ

    **Response:** Thank you for this question. This point is actually reflected in Fig. 6 where under gaining conditions, the warm surface water is prevented from going deeper into the sediment; whereas under losing conditions, the warm surface water infiltrates into the deeper sediment domain.

**R1_37** Therefore, in summer when river temperature is relatively high, the hydraulic conductivity is enhanced and becomes the main modulator for hyporheic exchange rate under losing condition.

- o Change the therefore language. The authors use this word a lot

    **Response:** Thank you for the suggestion. "Therefore" is replaced by "Consequently" (line: 342).

**R1_38** Combine the paragraphs between Lines 310 and 320

**Response:** Changed as suggested.

**R1_39** Line 343, Therefore, hyporheic zones have a larger cooling effect during high river temperature under out-of-phase gaining conditions than under in-phase conditions (under gaining conditions)

> **Response**: Added as suggested (line: 372).

**R1_40** Too many conditions maybe think of different wording for in-phase and out-of-phase (conditions)

> **Response:** We replaced a couple of "conditions" with "scenarios".

**R1_41** Loosing conditions speeds up residence time (RSF = reaction scale factor)

> **Response:** The reaction significance factor is proportional to the residence time (equation 8). $RSF_a$ under gaining conditions is around three orders of magnitude higher than under losing conditions due to the significantly longer residence time resulting from mixing between surface water and groundwater under gaining conditions.

**R1_42** Gaining conditions slows down residence time and allows mixing of GW and SW

> **Response:** Groundwater has significantly longer residence time. The mixing between groundwater and surface water under gaining conditions thus increases the mean residence time of the exfiltrating hyporheic fluxes. Therefore, $RSF_a$ under gaining conditions is higher than under losing conditions.

**R1_43** In conclusion, the timing 365 of groundwater table drawdown is more important under gaining conditions than under losing conditions for denitrification reactions.

> **Response:** Yes. With groundwater losing conditions, even though $RSF_a$ display peaks on a logarithmic scale, the actual differences of $RSF_a$ (in the scale of 10 to the power of $-5$) between in-phase and out-of-phase conditions are insignificant compared to gaining conditions.

**R1_44** Line 668 – could you mention this fact earlier in the paper, so the reader is not thinking about denitrification the entire time?

> **Response:** Thank you for the suggestion. The scaling of RSF for different reactions will be explained as soon as the RSF is first introduced in line 276:
>
> *It is worth noticing that instead of the denitrification, reaction potential of a different geochemical process can be assessed if a different characteristic time scale is applied in equation 8.*

**R1_45** Study limitations?

> o What about connectivity? A reference to some of this great work would be nice to see in this paper
>
> **Response:** Thank you for the suggestion. We added a short paragraph discussing hydrological connectivity at the end of discussion. Please refer to the specific comment **R1_3**.

**R1_46** Conclusion

> o Not strong enough or long enough
> o Need more space and references to specific aquatic community impacts and groundwater table diel drawdown.
>
> **Response:** Thank you for the comment. The conclusion was rephrased as suggested. Please find the changes made in line 461, 465, 466, and 469.

**Response to Comments from Referee #2**

The authors present an extensive modelling study on the interplay between diurnal temperature effects and groundwater gradients on the dynamic evolution of the hyporheic zone in a river with a defined bedform topography. The hyporheic zone is a highly relevant transition zone controlling biogeochemical processes such as denitrification in streams (e.g., Gomez et al. (2015)). Therefore, the topic of the manuscript fits well with the scope of HESS.

The processes affecting the exchange between river water, the hyporheic zone and groundwater are highly non-linear and can lead to seemingly counter-intuitive effects. The authors build on previous work (e.g., Wu et al., (2020, 2018)) and a model to investigate the questions specific to this manuscript. In particular, they study how daily temperature fluctuations in a stream impact the hyporheic exchange and how it interfers with effects caused by dial fluctuations of groundwater fluxes caused by evapotranspiration or pumping.

The authors provide a broad range of data and results on the hyporheic water fluxes, temperature gradients and potential impacts on biochemical process rates such as denitrification.

The manuscript is interesting. But before it can be published I suggest major revisions for clarifying open issues and for improving the structure to enhance readability.

> **Response:** Thank you for providing such a comprehensive and thoughtful review on the manuscript. The authors appreciate the detailed comments and suggestions. Below please find the point-by-point reply to the comments. Changes that in the manuscript are indicated by *underlined text in italic*. Line numbers in this response refer to the numbers in the track-changed manuscript. Based on the comments and suggestions presented by the reviewer, we modified the structure of the manuscript, both for single paragraphs and entire section; additional explanations and figures were added to clarify ambiguities and uncertainties.

**Major issues:**

R2_1    **Improve readability** The structure of the text is not always very reader-friendly. This means that it is not always easy to immediately understand and follow the logic of the

arguments and results. This observation holds true for single paragraphs as well as for entire sections (e.g., the Result section). Often the starting point of an argument is not what is directly evident to the non-specialists but the necessary explanations follow only afterwards.

The text on L. 55 - 58 may serve as an illustrative example: The starting point is that there are diel fluctuations of hyporheic exchange and that they may interact with diurnal changes of groundwater fluxes. However, for the non-specialist regarding the hyporheic zone, the diel fluctions may not be evident. Hence, upon reading one stops and reflects why this should be the case. In the current manuscript, the explanation comes only afterwards. I suggest a different structure:

1. Daily temperature fluctuations in stream (every reader will know and agree)
2. This affects viscosity and hence hydraulic conductivity (the readers will follow)
3. This induces diurnal changes in hyporheic exchange as demonstrated in Wu et al. (2020) (the reader will believe this)
4. There are also diel fluctuations in groundwater fluxes for several reasons (readers will know and agree)
5. Therefore, there are two dynamic processes affecting the hyporheic zone and they may potentially interact in rather non-linear ways.

This is just an example but I suggest to pay due attention to this aspect because the authors claim (with good reasons) that hyporheic processes have wider implications. This means their paper should also be read by a wider audience in the hydrology and water resources management community. Accordingly, they should write the paper for such an audience and consider what to expect from such readers as starting points for presenting the arguments and results.

> **Response:** Thank you for suggesting a very clear outline for modifying the text. Indeed the ideas can be conveyed much more clearly with the suggested structure. We modified L.53-63 following the suggested outline as below:
>
> *River temperature often fluctuates with a clear daily cycle in response to the diurnal change in solar radiation (Caissie, 2006). This daily change in river*

*temperature directly affects water viscosity and density, and subsequently the hydraulic conductivity of the sediment. As a consequence, hyporheic exchange rates often exhibit a diel fluctuation pattern due to the temperature-dependent hydraulic conductivity that governs the flow transport in the sediment. Wu et al. (2020) observe that hyporheic exchange fluxes inherit the daily-scale spectral signatures from river temperature fluctuations, and noticeably, however, these signatures are absent in river discharge of the studied site. This observation evidently indicates a strong control of the diel river temperature fluctuation on hyporheic exchange processes. However, the temperature-dependent diel rhythm of hyporheic exchange rates can be interfered by the daily groundwater table fluctuations due to evapotranspiration and anthropogenic pumping activities. Therefore, understanding the two players, namely daily groundwater hydraulic gradient change (as a result of daily groundwater table fluctuations) and diel hydraulic conductivity change (as a result of diel river temperature fluctuation), is important to characterize dynamic hyporheic exchange processes.*

R2_2 **Model description** There are several aspects of the model and its set-up that are not fully satisfactory:

1. *Model dimensions.* Given that the authors have used a 2-D model (L. 81), the model domain has to have dimensions along the x- and z-axes. Please provide this information (e.g., in terms of $\lambda$). Please demonstrate as well that this model set-up is a meaningful representation for the case study that represents a given real situation.

   **Response:** Thank you for this suggestion. We have added: *The streamwise length and the depth of the modeling domain are L = 3$\lambda$ and $d_{gw}$ = 5$\lambda$, respectively.(added in Line 79)*

   To demonstrate if the model set-up is a meaningful representation, the following paragraph is added in section 4.5 "Study Limitation" from line 448:

*The morphological setting of the model is dune with aspect ratio of 0.1 under subcritical flow conditions with a Froude number around 0.39 (Bridge, 2009; Dingman, 2009). The geological setting has been simplified as homogeneous and isotropic porous media. Even though the sediment in nature can rarely be homogeneous and isotropic, this simplification is necessary for improving computational efficiency without defeating the objective of identifying the interactions among river discharge, temperature and groundwater dynamics.*

2. *Fig. 2.* At that point, the panels b and c are rather confusing. Panel a is very generic, but on the lower panels real dates are given and it is not clear to the reader what these values on the x-axes mean and why the are chosen. It is also obscure what the temperature represents. It takes a lot of reading until one can make the link to the case study and the respective observations.

> **Response:** Thank you for pointing out this issue. The dates in x-axes were chosen randomly with the objective of presenting the difference between the in-phase and out-of-phase scenarios. Because the groundwater flux was conceptualized as uniform sinusoidal curve, plotting it for a long period would make these two scenarios hard to distinguish. After plot experimenting, a 10-day time window is appropriate to preserve the difference between the two scenarios. To clarify the meaning of the x-axes, the following sentences are added in the figure caption:
>
> *Temperature time series are obtained from the U.S. Geological Survey (USGS, Site ID: 06893970). Groundwater flux is conceptualized as sinusoidal curves with varying amplitudes representing the strength of the groundwater upwelling or downwelling, and varying phases representing in-phase and out-out-phase scenarios. For figure clarity, a 10-day time window is selected arbitrarily from Jun 21 to Jun 30, 2017.*

3. Mass balance. From Fig. 2 (a), it follows that the water balance for the model domain is given by $Q_{river-out}$ (t) = $Q_{river-in}$ (t) + $q_b$ (t). Based on how the boundary conditions are defined however, the water flow in the river is independent on the groundwater fluxes imposed (the flow simply follows from the prescribed $H_s$ (t) (Eq. 2, 3). Also the head distribution at the water-sediment interface is flux-independent. However, this distribution was derived from empirical observations Elliott & Brooks (1997) without considering gaining or losing situations. This seems to be adequate as long as $U_s$ (t) $H_s$ (t) >> $q_b$ (t) $L_{domain}$ with $L_{domain}$ being the length of model domain. Please i) provide the evidence that this holds true for the case study and the dimension of the model domain, and ii) make these aspect also clear in the discussion. Actually, this aspect seems to emphasis the importance of the findings: even small groundwater fluxes may have a pronounced influence on the hyporheic zone. This may be evident to the authors, but I missed that point in the context of the entire paper.

> **Response:** This is a good point. We calculated the river discharge and groundwater discharge/recharge as the reviewer suggested. The results indicate that the river discharge is 4 orders of magnitude higher than the groundwater discharge/recharge (Figure R1), suggesting that ignoring the impact of groundwater flow on the head distribution at the sediment-water interface is a reasonable simplification. To address this issue in the manuscript, the following sentences are added in the Discussion 4.5 "Study Limitation" (line: 442):

> *Additionally, in the present study the surface water flow is assumed as an independent system that is not affected by groundwater flows. This simplification can only be used when groundwater discharge or recharge is significantly smaller than the river discharge. In our case, the groundwater discharge or recharge is at least 4 orders of magnitude lower than the river discharge. Therefore, this*

*simplification has limited impact on the results. The noticeable difference in the magnitude between groundwater discharge/recharge and river discharge also emphasizes the finding that even small groundwater fluxes may have a pronounced influence on the hyporheic zone.*

[Figure]

Figure R1: River discharge and groundwater discharge/recharge in logarithmic scale ($\log_{10}[m^3/s]$). River discharge time series are obtained from the U.S. Geological Survey (USGS) with the site no. 06893970 and observations in the year of 2015. Groundwater flux is conceptualized as sinusoidal curves with varying amplitude and phases. The groundwater flux presented here is with the highest amplitude among the three scenarios explored. The groundwater discharge or recharge is at least 4 orders of magnitude lower than the river discharge. Therefore, ignoring the impact of groundwater discharge on the surface water flow will not affect the results. Note that this figure is only used for review purpose.

4. *Eq. 6a.* I could not find an explanation for $a_0$. It is tedious to go to previous publications and guess that $a_0 = 1$.

**Response:** Thank you for pointing out this problem. The following sentence is added in line 124:

*"the initial condition for the moments $a_0 = 1,...$"*

5. *Model implementation.* Please provide some information on the model implementation (grid set-up, model version, run time etc.).

> **Response:** Thank you for this suggestion. The following information is added at the end of Method section line 80:
>
> *The flow and transport models described are solved with the finite element method implemented in COMSOL Multiphysics (version: 5.4) using a mesh with telescopic refinement near the boundaries and approximately 54,000 elements. The computation time for a full-length scenario is around 60 hours.*

6. *Defining the hyporheic zone.* It is unclear how the procedure described on L. 130 - 136 is actually implemented. First, because the hyporheic zone changes over time, the proposed procedure needs to be repeated, I assume. Can you comment on that? Second, for neutral and losing conditions, it seems that the threshold $C \geq 0.9C_s$ will eventually exceeded across the entire domain. Can you clarify?

   **Response:** Thank you for this question. *The boundary of the hyporheic zone is renewed at every time point with the threshold $C \geq 0.9C_s$. Therefore, the boundary of the hyporheic zone is changing over time under varying flow conditions.*(added in line 134)

   For neutral condition, we think that the threshold might not be exceeded eventually because of the underflow (or baseflow) driven by the horizontal pressure gradient induced by the channel slope. This horizontal pressure can limit the hyporheic zone expansion under rising hydraulic gradient at the streambed. For losing conditions, it is true that the threshold will be eventually exceeded across the entire domain. Therefore, it is common to use reversed Darcy flow to define the hyporheic zone under losing

conditions in order to track the subsurface regions that are really flushed by surface water. The results presented in the manuscript were not based on flow-reversed losing condition simulations. To find out the difference, we have re-run all the losing scenarios with flow-reverse. The results of exfiltrating hyporheic fluxes, temperature of exfiltrating hyporheic fluxes, and mean residence time distributions show nearly no differences compared with the results simulated without flow-reverse. Only the infiltrating hyporheic fluxes show higher fluctuation amplitudes. The figure below is for the same metrics as Figure 5 in the manuscript but with simulated hyporheic fluxes using flow-reverse (Figure R2).

[Figure]

Figure R2: This is the same figure as Figure 5 in the manuscript but with simulations using flow reverse.

To our understanding, if strictly following the definitions from Triska et al. (1989) and Gooseff (2010), tracking HZs with flow reverse is not necessary for losing conditions. However, after some discussions we think that tracking HZs under losing conditions using flow-reverse is more appropriate to identify the areas with the largest influence from the surface

water. Therefore, we added more details for tracking HZ under losing conditions in the method section (Line 135):

*With this condition, the threshold $C \geq 0.9C_s$ will be eventually exceeded across the entire domain under losing conditions. Therefore, hyporheic zone is tracked using reversed Darcy flow in order to identify the areas with the largest influence from the surface water under losing conditions.*

Additionally, the Figure 5 has been replaced with the simulations results using reversed flow field as shown here in Figure R2. Changes are made in the Results description for losing conditions in line 244 and the Discussion in line 292.

R2_3 **Description of the case study** This description is very superficial and has to be improved substantially.

1. *Site identification and description* Please provide more information on the site including the location and name. It is not necessary that every interested reader has to check the USGS website. Describe some key characteristics of the climate and hydrology of the catchment and the measuring site (altitude, mean discharge etc.). This is important to put the findings in a proper context.

   It is also essential to know which observation period was used for the simulations. One learns only at a later stage (e.g., from Fig. 3a) that three hydrological years seem to have been used.

   **Response:** Thank you for this suggestion. The following site description is added in Line 142:

   *The gauging station is located in Spring Branch Creek at Holke Road in Independence, Missouri (ID: 06893970, Lat 39°05'18", Long 94°20'36" referenced to North American Datum of 1927). The station is on upstream left bank Missouri Highway 78 about 2.4 km above the confluence with the*

*Little Blue River with a drainage area of 22 km². The observation period is from 2014-10-16 to 2017-10-16.*

On L. 160, the amplitude of groundwater flux changes are linked to a range of the groundwater table fluctuations. Although a reference is provided, this is not sufficient. Boano et al. (2008) presents a general framework for linking stream-groundwater interactions and the influence on the hyporheic zone, but not any site-specific information for this case study. Describe the approach including the equations used and the model assumptions. In this context, it would be also useful to provide evidence that this assumed water table fluctuation is also reasonable for a hypothetical groundwater pumping operation.

> **Response:** Thank you for the suggestion. To better describe the method referred, the following paragraphs are added in the Supplementary Information:
>
> *Boano et al. (2008) performed a number of simulations for different stream aspect ratios (the ratio between river half-width and river stage) and average slopes of the groundwater table, and found out that the upwelling velocity has a linear correlation with the slope of the groundwater table:*
>
> $$\frac{q_b}{K} = 0.57 \frac{dh}{L_w}$$
>
> *Where $q_b$ is the groundwater upwelling velocity, $K$ is the hydraulic conductivity which is $10^{-3}$ m/s in this study, dh is the head difference between river stage and groundwater table elevation, $L_w$ is the half-width of the river channel which is 2.5 m.*
>
> *In the present study, we made use of this linear relationship to evaluate how much the head difference dh would change due to the daily groundwater level fluctuations. To achieve this objective, we made additional assumptions that the distance between the river bank and the hypothetical groundwater level observation point is equal to the river half-width, $L_w$; and the slope of the groundwater table is less than 0.1. The average river aspect ratio in the model setting is around 25, which falls within the range of the explored aspect ratios in Boano et al. (2008).*

*With the highest groundwater level fluctuation amplitude, $q_b$ varies daily from $1 \times 10^{-6}$ m/s to $9 \times 10^{-6}$ m/s, resulting in a change in the head difference dh of 3.5 cm. With the medium groundwater level fluctuation amplitude, the change in the head difference dh is 1.8 cm. With the lowest groundwater level fluctuation amplitude, the change in the head difference dh is 0.9 cm.*

In the manuscript, the following text was added in line 172:

*Using the method proposed in Boano et al. (2008) which is described with details in Supplementary Information, a change in the head difference (dh) of 3.5 cm is observed with the highest groundwater level fluctuation amplitude where qb varies daily from $1 \times 10^{-3}$ m/s to $9 \times 10^{-3}$ m/s. With the medium groundwater level fluctuation amplitude, the change in the head difference dh is 1.8 cm. With the lowest groundwater level fluctuation amplitude, the change in the head difference dh is 0.9 cm.*

The paragraph on L. 144 - 155 describes the *in-phase* and *out-of-phase* conditions. It might enhance the intuitive understanding for a general reader if the authors indicate more explicitly that the *out-of-phase* conditions represent the natural state with high stream temperatures and lower water table in the aquifer due to transpiration by the vegetation.

**Response:** Thank you for the suggestion. The following text is added in line 158:

*Under gaining scenarios, out-of-phase conditions represent the natural state that highest air/river temperature occurs at the lowest water table (resulting to lowest groundwater flow rate) in the aquifer due to transpiration by the vegetation; under losing scenarios, in-phase condition represents the natural transpiration condition, because the highest air/river temperature contributes to the strongest transpiration which results in a larger hydraulic head difference between river and aquifer, and thus contributes to the higher losing groundwater fluxes.*

R2_4   **Result section:** This section contains a lot of material (which is positive) but the way of presenting needs improvement. The more so because not all of the necessary results seem to be shown so far.

1. *Structure* One of the key messages of the manuscript is that there is an intricate interplay between the temperature regime, the flow regime of the stream and the water table fluctuations in the aquifer that needs to be understood. To be able to understand this, one has to get an overview about the general conditions prevaling at the study site during the period of interest. Therefore, I suggest to start with a short description of the key features of the three hydrological years.

   Subsequently, it helps the reader if the complexity is increased in a stepwise fashion. Therefore, I would first describe the results for the neutral conditions, then the losing conditions and finally the gaining conditions. Furthermore, I suggest to use explanations such as on L. 277 - 279 to frame the result section in a way that is intuitive also to the non-specialist reader.

   > **Response:** Thank you for this good suggestion. The results section is re-organized with progressively increased complexity by first introducing the neutral conditions, and then followed by loosing and gaining conditions. Please find the changes made for the results section in the track-changed document. To include a short description of the key features of the three hydrological years, the following text is added in line 187:
   >
   > *In the observation period, the river discharge is intermittent and characterized by short recession periods (approximately from 2 to 1500 $m^3$/s); the river temperature shows clear seasonal variations (approximately from 0 to 35°C) and daily fluctuation. Mean annual precipitation at the gauge location is 106 cm. Average annual air temperature at the gauge location is 12.6 °C. There is no dam in the watershed.*

2. *Nomenclature* One of the confusing things is the terminology used for describing the hyporheic fluxes. Nowhere it is explained what actually meant by the

infiltrating and exfiltrating hyporheic fluxes. For the neutral case, the two fluxes are identical, which makes sense. Under gaining conditions, the infiltrating flux is consistently larger than the exfiltrating flux. How is this explained and why is the same true for the losing conditions when there is a net flux from the river to the aquifer? Please clearly define the terms and explain the apparent contradictions mentioned.

> **Response:** Thank you for asking this question. We have added: *Using this definition, water flow into the hyporheic zone is defined as infiltrating hyporheic fluxes and water flow out of the hyporheic zone is defined as the exfiltrating hyporheic fluxes.* (Line 137)
>
> For the neutral case, even though the differences are trivial, we think the two fluxes are not identical due to the temperature-dependent fluid properties. If the geochemical definition of hyporheic zone is applied as in this case, these two fluxes will also be different due to the hyporheic zone boundary delineation. Under gaining conditions, the exfiltrating hyporheic fluxes show enhanced fluctuation amplitudes compared with the infiltrating hyporheic fluxes due to the additional fluctuations in the gaining groundwater fluxes that are mixed with the hyporheic fluxes which originate from the surface. Under losing conditions, since we reversed the flow directions when tracking hyporheic zone as discussed in the response to the comment "*6. Defining the hyporheic zone*" under **R2_2**, the results are different than that presented in the manuscript. As indicated in the Figure R2, the infiltrating hyporheic fluxes have higher fluctuation amplitudes because there is no mixing in the exfiltrating hyporheic fluxes under losing condition as the mixing occurred under gaining conditions according to the geochemical definitions of hyporheic zones.

3. *Residence times* The method sections describe how to estimate time variable residence times in the hyporheic zone. Despite of using an average value for calculating the reaction significance factor RSF, no data on residence times are provided. This is essential if one would like to be able to evaluate the relevance of the results for any biological or bio(geo)chemical processes. Provide the results on

the time-variant residence times and how they change upon the different boundary conditions.

> **Response:** Thank you for this suggestion. The calculation of RSF is explained in the response to the comment below. Mean residence time plots were added in Fig. S2 in the Supplementary Information.

4. *RSF* First of all, this approach has not been introduced so far. It should be mentioned in the Introduction when introducing the denitrification topic and described in the method section. Apart from that I am not sure whether the chosen form is an adequate implementation of the concept. I have three question marks:

   (a) The first relates to $q_{HZ}$ because I could not follow what this term actually represents (see above: how does it relate to infiltrating and exfiltrating fluxes?).

   > **Response:** Please refer to the response to the comment above.

   (b) Why is the mean residence time used for calculating a time-variant quantity such as RSF when residence times were derived as a function of time? Depending on the temporal correlation functions between the relevant hyporheic flux $q_{HZ}$ and the residence times $\tau_{HZ}$, there might be substantial deviations from the current version.

   > **Response:** Thank you for asking this important question. To be more precise, it is the mean of the probability distribution of the residence time in any given time point. To clarify the meaning of $\tau_{HZ}$, the following sentence is added in Line 272:
   >
   > *$\tau_{HZ}$ is the mean of the probability distribution of the residence time at any time point [T].*

   (c) The time scales of denitrification. First, the description of how $\tau_{HZ}$ was parameterised is insufficient. Which quantiles in Gomez et al. (2015) do you refer to? Second, denitrification depends very much on temperature (e.g., Boulêtreau et al. (2012)). This implies that $\tau_{dn}$ is not constant. Given that the manuscript deals with temperature as a key influencing factor, it would seem

logic to consider such a temperature dependence also for $\tau_{dn}$. At least one could test the sensitivity of RSF against the temperature dependence of denitrification.

> **Response:** Thank you for these suggestions. Firstly, to better present the values of $\tau_{dn}$ we added a new figure in the supplementary information to show the quantiles of the characteristic time scales for denitrification. Please find the details in the response to the second comment from reviewer #1 (R**1**_**2**) .
>
> The second suggestion is also a good point. We add the following text to clarify in the manuscript in Line 405:
>
> *The temperature-dependence for $\tau_{dn}$ is not considered, however we use both the 25th and 75th quantiles as the lower and upper ranges for calculating RSF, which mostly include the variations caused by the changing temperature as indicated in Zheng et al. (2016) where a roughly five-fold increase was observed in denitrification rates when temperature increased from 5 °C to 35 °C.*
>
> The reviewer's suggestion is a great idea. We consider to test it with a model including temperature-dependent denitrification processes. However, this is beyond the scope of the present manuscript and hopefully we can present it in a future manuscript at a later date.

5. *Plausibility check against empirical data* One of the values of such a model study is the possibility to study processes and their interactions under well defined conditions and to explore system behaviours that are otherwise impossible to obtain. This comes at the costs of the difficulty to relate the model findings and insights back to the real world. To improve on that the authors should provide more context on the case study (see above). On the other hand, they should also add some comparisons of model results with empirical observations to provide some plausibility checks. Possibilities for doing so would for example be the extent of the hyporheic zone, residence times (both not even shown for the model

results, see above) or RSF values as depicted in Fig. 8. Such values could for example be compared to estimates provided by Gomez et al. (2015).

> **Response:** We agree with the reviewer. The dimensionless RSF serves as an appropriate metric that can be used for comparisons with the other observations. Note that Gomez et al. (2015) only presented total RSF for denitrification (given by the sum of the vertical and lateral RSF), therefore we compared our results with Harvey et al. (2018) where RSF for riverbed induced hyporheic exchange was calculated. The following text is added in Line 395:
>
> *In Harvey et al. (2019), RSF was calculated with mean annual hyporheic flux and river discharge without considerations of the temporal variability of the flow conditions and groundwater upwelling/downwelling. To be able to compare with the results, we also calculated mean RSF using mean river discharge and mean hyporheic fluxes. The calculated mean RSF is approximately from -2.7 to -1.8 for gaining condition and -5.8 to -4.8 for losing condition, which roughly falls within the range of the mean RSF observed in Harvey et al. (2019). Under loginsg condition, the RSF is smaller than the values reported in Harvey et al. (2018), because losing conditions significantly reduce the denitrification potential as indicated in Fig. 8.*

**Detailed comments:**

**L. 18 - 19:** Why is this understanding *key to water resources management*? There are many aspects relevant for water management (land use management, hydropower generation schemes etc.). Please be more specific for aspects this understanding is key and why.

**Response:** Thank you for this question. This sentence is rephrased as below (line 19)

*Understanding the spatiotemporal variability of hyporheic exchange processes is key to characterizing the nutrient cycling and river ecosystem functioning (Lewandowski et al., 2019)*

**L. 23, 26 and elsewhere:** Articles or pronouns are missing sometimes. Please have a linguistic check.s

**Response:** Done as suggested.

**Fig. 4:** Explain the time axes and give a reason why only that part of the entire study period is displayed? It seems to be rather arbitrary. Are the results from the *in-phase* or *out-of-phase* simulations?

**Response:** The results are selected arbitrarily with the considerations of figure clarity. 10-day time window was selected, because longer time window makes the plots hard to distinguish. Effects of groundwater table fluctuation amplitudes on dynamic hyporheic responses are only explored under in-phase scenarios, because under out-of-phase scenarios, fluctuations of exfiltrating hyporheic fluxes are almost always in the same phase with the diel river temperature fluctuations. Therefore, unlike in-phase scenarios, the phase shifts due to reduced amplitudes in groundwater table fluctuation are not observed. Reduced amplitudes in groundwater table fluctuation under out-of-phase scenarios only contribute to reduced amplitudes in exfiltrating hyporheic flux fluctuations. For simplicity, only results in in-phase scenarios are presented (Line 230-235).

**Fig. 6:** Unfortunately, one can hardly see the differences between *a* and *b* or *c* and *d*, respectively. One option could be to show the respective difference plots and to add difference plots for the fluxes.

**Response:** Thank you for pointing out this issue. We only kept 2 sub-figures to illustrate the differences of heat transport in gaining and losing conditions. The differences between in-phase and out-of-phase scenarios are explained better with Fig. 7.

**Fig. 8:** Add the year to the time axes and explain why this specific period was selected.

**Response:** The year 2017 is added. The results are selected arbitrarily with the considerations of figure clarity.

**References**

Boano, F., R. Revelli, and L. Ridolfi. 2008. Reduction of the hyporheic zone volume due to the stream-aquifer interaction. Geophysical Research Letters 35.

Boulêtreau, S., E. Salvo, E. Lyautey, S. Mastrorillo, and F. Garabetian. 2012. Temperature dependence of denitrification in phototrophic river biofilms. Science of the Total Environment 416:323-328.

Boulton, A. J. (2007). Hyporheic rehabilitation in rivers: restoring vertical connectivity. Freshwater Biology, 52(4), 632-650.

Bridge, J. S. (2009). Rivers and floodplains: Forms, processes, and sedimentary record. John Wiley & Sons.

Butler, J. J., Zlotnik, V. A., & Tsou, M. S. (2001). Drawdown and stream depletion produced by pumping in the vicinity of a finite width stream of shallow penetration. Ground Water, 39(5), 651-659.

Butler, J. J., Zhan, X., & Zlotnik, V. A. (2007). Pumping-induced drawdown and stream depletion in a leaky aquifer system. Groundwater, 45(2), 178-186.

Caissie, D. (2006). The thermal regime of rivers: A review. Freshwater biology, 51(8), 1389–1406.

Dingman, S. L. (2009). Fluvial hydraulics. Oxford; New York: Oxford University Press.

Elliott, A. H., and N. H. Brooks. 1997. Transfer of nonsorbing solutes to a streambed with bed forms: Theory. Water Resources Research 33:123-136.

Gomez-Velez, J. D., J. W. Harvey, M. B. Cardenas, and B. Kiel. 2015. Denitrification in the Mississippi River network controlled by flow through river bedforms. Nature Geoscience 8:941- 945.s

Gooseff, M. N. (2010). Defining hyporheic zones–advancing our conceptual and operational definitions of where stream water and groundwater meet. Geography Compass, 4(8), 945–955.

Harvey, J., Gomez-Velez, J., Schmadel, N., Scott, D., Boyer, E., Alexander, R., ... & Moore, R. (2019). How hydrologic connectivity regulates water quality in river corridors. JAWRA Journal of the American Water Resources Association, 55(2), 369-381.

Lewandowski, J., Arnon, S., Banks, E., Batelaan, O., Betterle, A., Broecker, T., ...& Gomez-Velez, J.: Is the hyporheic zone relevant beyond the scientific community? Water, 11(11), 2230, 2019.

Malard, F., Tockner, K., DOLE-OLIVIER, M. J., & Ward, J. V. (2002). A landscape perspective of surface–subsurface hydrological exchanges in river corridors. Freshwater Biology, 47(4), 621-640.

Roley, S. S., Tank, J. L., & Williams, M. A. (2012). Hydrologic connectivity increases denitrification in the hyporheic zone and restored floodplains of an agricultural stream. Journal of Geophysical Research: Biogeosciences, 117(G3).

Stanford, J. A., & Ward, J. V. (1993). An ecosystem perspective of alluvial rivers: connectivity and the hyporheic corridor. Journal of the North American Benthological Society, 12(1), 48-60.

Triska, F. J., Kennedy, V. C., Avanzino, R. J., Zellweger, G. W., & Bencala, K. E. (1989). Retention and transport of nutrients in a third-order stream in northwestern California: Hyporheic processes. Ecology, 70, 1893–1905.

Ward, J. V., Malard, F., Stanford, J. A., & Gonser, T. (2000). Interstitial aquatic fauna of shallow unconsolidated sediments, particularly hyporheic biotopes. In 'Subterranean Ecosystems'. (Eds H. Wilkens, DC Culver and WF Humphreys.) pp. 41–58.

Wu, L., J. D. Gomez-Velez, S. Krause, T. Singh, A. Wörman, and J. Lewandowski. 2020. Impact of Flow Alteration and Temperature Variability on Hyporheic Exchange. Water Resources Research 56:e2019WR026225.

Wu, L., T. Singh, J. Gomez-Velez, G. Nützmann, A. Wörman, S. Krause, and J. Lewandowski. 2018. Impact of Dynamically Changing Discharge on Hyporheic Exchange Processes

Under Gaining and Losing Groundwater Conditions. Water Resources Research 54:10,076- 010,093.

Zheng, L., M. B. Cardenas, and L. Wang (2016), Temperature effects on nitrogen cycling and nitrate removal-production efficiency in bed form-induced hyporheic zones, J. Geophys. Res. Biogeosci., 121, 1086–1103, doi:10.1002/2015JG003162.

---

## Referee Report (RR1)

The authors have investigated an interesting aspect of hyporheic exchange that has yet to receive much attention. Although the results are compelling, I found that there is a large discrepancy between what was simulated and its conceptual interpretation. The issue with the conceptual interpretation of the model results is that the prescribed flux boundary along the bottom of the model domain does not directly represent groundwater table drawdown at some distance away from the river. My comments below should clarify this discrepancy.

Detailed comments:

1. The modelling does not actually simulate the effects of daily groundwater table drawdown.

The boundary condition that conceptually represents a daily fluctuation in groundwater table is a prescribed flux on the bottom boundary of a 2D model. The title should instead read: "How does dynamic losing and gaining river conditions affect the diel rhythm of hyporheic exchange?"

There is a disconnect between the way the authors describe the groundwater table and the conditions at the river. Only in lines 428 to 433 do the authors explain this conceptually. It is possible that pumping can lower groundwater tables, thus effecting the hydrogeologic gradients and fluxes to a river. However, this depends on the hydraulic properties between the pumping well and the stream (transmissivity, storativity), as well as the distance between the pumping well and the stream (Barlow and Leake, 2012). Unless the well is directly connected and next to the river, there will be a delay between the start of pumping and its influence on hydrogeologic conditions at the river (i.e., magnitude of river gaining/losing).

How realistic is the prescribed flux boundary that represents daily groundwater withdrawals? I believe that this boundary condition is not realistic in cases where the pumping well is far away from or is not well connected to the river. Hence, I would remove from the manuscript any mention of groundwater table dynamics and simply discuss changing in river losing/gaining conditions, which is what was simulated. Alternatively, more should be stated at the beginning of the manuscript about the assumptions the authors have made about hydraulic connection between the dewatering well and river, as well as the distance between said well and river (i.e., they need to be relatively close to one another). I have referenced specific lines below with where I think these changes should be made. There are many more places in the manuscript where this wording should be changed, I have only listed a few.

Line 68 – The aim to quantify the impact of "groundwater table drawdown" on hyporheic exchange processes. Replace "groundwater table drawdown" with "the degree of gaining/losing conditions".

Fig 3 – "Effect of diel river temperature fluctuations and daily groundwater table drawdowns on hyporheic fluxes…". Again daily groundwater table drawdown was not simulated, but daily fluctuations in gaining/losing conditions of the river.

Line 300 – "The timing of groundwater table drawdown also affects hyporheic exchange rates." Upward and downward fluxes representing gaining/losing conditions affects hyporheic exchange rates. There is a conceptual disconnect here.

Line 310 – "Therefore, the timing of the aquifer pumping can potentially amplify or reduce the dispersal of pollutants in the aquifer". It would be better to state that the 'timing of river gaining/losing magnitude can potentially amplify or reduce…"

Lines 313 to 317 - Sure, but it also depends on the level of hydraulic connection between the river and the well (Barlow and Leake, 2012).

Line 380 – Scheduling pumping activities to protect thermal heterogeneity across multiple spatial scales again depends on the well-stream connection, if any.

Line 392 – "hyporheic denitrification potential can be regulated by adjusting the timing of daily groundwater table drawdown." It would be better to state the "timing of hydraulic gradients towards the river or groundwater in/out flow to the river" and not "timing of daily groundwater table drawdown". The influence of groundwater table drawdown on the state of the river depends on its hydraulic connection and distance between the well and the river. It's possible to have a delayed response in the rivers condition or no response at all if there is little to no connection or if the well is very far from the river.

Line 456 – "Groundwater table dynamics" replace with "Groundwater discharge/recharge to/from rivers substantially…"

Line 471 – "…hyporheic denitrification potential is also changing following groundwater table drawdown." Replace "groundwater table drawdown" with "groundwater discharge/recharge". For the statement made be the authors about groundwater tables, one would require a 3D model that contains areas beyond the river banks. Instead, what has been simulated is a 2D section along the river with a prescribed flux boundary representing fluctuating gaining/losing conditions. There is a conceptual gap between the prescribed boundary condition simulated in this study and groundwater table drawdowns that the authors describe. Groundwater table drawdowns could be happening at some distance away from the river with varying hydraulic properties between, which would dramatically influence the degree to which aquifer pumping would effect gaining/losing conditions at the river.

2. Model parameters

I am trying to gauge the realism of the model parameters but am having a hard time finding certain values. I have listed the specific line numbers for parameter values that were not stated or difficult to find. Clearly stating the parameter values would help future researchers to extend or repeat these numerical experiments.

Line 91 – What were the values selected for permeability and porosity?

Line 100 – Do you have references to support this aspect ratio?

Line 108 – What is the hydrodynamic thermal dispersion tensor value chosen?

Line 288 – "mainly due to the change of hydraulic conductivity which is a function of diel temperature fluctuations." Again what was the permeability value chosen? How much does hydraulic conductivity vary in your simulations due to temperature? Line 341 states that there is a 220% change in K due to a 30 degree change in temperature, but I'm not sure if this is a realistic magnitude of daily temperature change or K change. Thomas (2014) shows temperature fluctuations over a year in Sauk River, Washington. There is a 18 degree C range over the year, but at the daily scale there is rarely more than a 10 degree fluctuation throughout the day. Perhaps the daily temperature range should be decreased by 3x, unless the authors have references that support a daily 30 degree temperature change in river water.

3. Other comments

Line 28 – Chow et al. (2019) conducted a sensitivity analysis on the effects of river bathymetry (i.e., geomorphological settings) on meander-scale hyporheic exchange.

Line 30 – Chow et al. (2020) evaluated sediment heterogeneity and its effects on meander-scale hyporheic exchange.

Line 40 – "Large groundwater upwelling and downwelling may compress 'or extend' hyporheic…"

Fig 1b and c – The hyporheic exchange should be compressed in Fig 1b and extended in Fig 1c. Instead it looks the same between Fig. 1b and c.

Lines 324 to 330 – I find this sentence confusing. So what plays a dominant role in the winter? Pumping? Please clarify.

Line 380 – Remove s from 'cares'

Line 387-389 – Wouldn't losing conditions have longer residence times since the flow paths would be extended and stretched?

Line 467 to 468 "The timing of aquifer pumping should be adjusted to avoid…". Can you be more specific here. I.e., "The pumping should decrease or stop during flood events in order to ensure minimal contaminant uptake".

Lastly, a general comment about the two scenarios of in-phase and out-of-phase compared throughout the manuscript. It would be nice to get from the authors their ideas on how likely these two scenarios are and what kinds of assumptions must be met in order for them to be realistic. I can imagine that in-phase is a more likely scenario because ET tends to increase as temperatures increase during the day. Also, in cases where there is hydraulic connection and a short distance between the well and river, I would expect that groundwater usage would increase following daytime activity. Out-of-phase may be less likely, but this could depend on the well-stream connection.

**References**

Barlow, P.M. and Leake, S.A., 2012. Streamflow depletion by wells: understanding and managing the effects of groundwater pumping on streamflow (p. 84). Reston, VA: US Geological Survey.

Chow, R., Bennett, J., Dugge, J., Wöhling, T. and Nowak, W., 2020. Evaluating subsurface parameterization to simulate hyporheic exchange: The Steinlach River Test Site. Groundwater, 58(1), pp.93-109.

Chow, R., Wu, H., Bennett, J.P., Dugge, J., Wöhling, T. and Nowak, W., 2019. Sensitivity of simulated hyporheic exchange to river bathymetry: The Steinlach River test site. Groundwater, 57(3), pp.378-391.

Thomas, L., 2014. Stream Temperature Variability: Why It Matters to Salmon. Science Findings, United States Department of Agriculture Forest Service. https://www.fs.fed.us/pnw/sciencef/scifi163.pdf

---

## Author Response (AR2)

Dear Dr. Stamm,

Thank you for the helpful comments and suggestions on our manuscript entitled "How does daily groundwater table drawdown affect the diel rhythm of hyporheic exchange?" [MS No.: hess-2020-288]. Below we provided a point-by-point reply to reviewer #3's comments. To summarize, the following major changes were made:

1. To clarify the model assumption on groundwater boundary conditions, we reworded the research objectives in Introduction section (line 69) and clarified the selection of groundwater scenarios in Method section (line 165).

2. Additional information is provided for model parameterizations.

We look forward to your further reply!

Kind regards,

Liwen Wu

c.c.: Jesus D. Gomez-Velez, Stefan Krause, Anders Wörman, Tanu Singh, Gunnar Nützmann, and Jörg Lewandowski

**Response to Comments from Reviewer #3**

**General Comments**

The authors have investigated an interesting aspect of hyporheic exchange that has yet to receive much attention. Although the results are compelling, I found that there is a large discrepancy between what was simulated and its conceptual interpretation. The issue with the conceptual interpretation of the model results is that the prescribed flux boundary along the bottom of the model domain does not directly represent groundwater table drawdown at some distance away from the river. My comments below should clarify this discrepancy.

> Response: The authors thank this reviewer's comments regarding the selection of groundwater boundary conditions and model parameterizations. The point-by-point reply to the comments is given below. Changes in the manuscript are indicated by *underlined text in italic*. Line numbers in this response refer to the numbers in the track-changed manuscript. By responding to the following comments, we clarified the model assumptions and included more information on the model parameterization.

**Detailed Comments:**

1. The modelling does not actually simulate the effects of daily groundwater table drawdown. The boundary condition that conceptually represents a daily fluctuation in groundwater table is a prescribed flux on the bottom boundary of a 2D model. The title should instead read: "How does dynamic losing and gaining river conditions affect the diel rhythm of hyporheic exchange?"
   There is a disconnect between the way the authors describe the groundwater table and the conditions at the river. Only in lines 428 to 433 do the authors explain this conceptually. It is possible that pumping can lower groundwater tables, thus effecting the hydrogeologic gradients and fluxes to a river. However, this depends on the hydraulic properties between the pumping well and the stream (transmissivity, storativity), as well as the distance between the pumping well and the stream (Barlow and Leake, 2012). Unless the well is

directly connected and next to the river, there will be a delay between the start of pumping and its influence on hydrogeologic conditions at the river (i.e., magnitude of river gaining/losing).

How realistic is the prescribed flux boundary that represents daily groundwater withdrawals? I believe that this boundary condition is not realistic in cases where the pumping well is far away from or is not well connected to the river. Hence, I would remove from the manuscript any mention of groundwater table dynamics and simply discuss changing in river losing/gaining conditions, which is what was simulated. Alternatively, more should be stated at the beginning of the manuscript about the assumptions the authors have made about hydraulic connection between the dewatering well and river, as well as the distance between said well and river (i.e., they need to be relatively close to one another). I have referenced specific lines below with where I think these changes should be made. There are many more places in the manuscript where this wording should be changed, I have only listed a few.

> Response: The reviewer has raised concerns on how the distance between well and river affects the groundwater level fluctuations. However, this is not directly relevant in this study with the proposed model settings and research scope.
>
> The model uses groundwater fluxes with daily fluctuations to study how groundwater dynamics caused by evapotranspiration or pumping impact the hyporheic exchange and how they interfere with the effects caused by daily temperature fluctuations in a river. A superimposed groundwater flux on the lower boundary offers an efficient way to investigate these complex interactions among river stage, river temperature and groundwater table fluctuations, and their impacts on hyporheic exchange. As discussed in line 429 to 433 in the study limitation, although the superimposed groundwater flux was not a perfect representation of the real groundwater response, this simplification allowed us to investigate hyporheic exchange dynamics under complicated multifactorial interactions efficiently. The hyporheic dynamics that cannot be captured by using superimposed groundwater fluxes were listed and discussed in line 433-442.

The groundwater fluxes time series are conceptualized as sinusoidal. In other words, the groundwater fluxes were not observed but synthetic datasets. There is hence no actual well that was measured. To state this point clearer, the research objective at the end of introduction was reworded as below (line 69):

*With these objectives in mind, a series of synthetic groundwater scenarios corresponding to different timings of groundwater table drawdown under gaining and losing conditions is applied in a physically based hyporheic flow and heat transport model.*

Additionally, the reviewer suggested to change the title with only mentioning "losing and gaining river conditions". However, the groundwater scenarios used in the study not only include different groundwater flow directions (gaining or losing), but also include the timing of drawdown (in-phase or out-of-phase) and the fluctuation amplitudes (low, medium and high). Therefore, only use "losing and gaining" cannot accurately describe the groundwater scenarios explored in this study. Therefore, we have kept the original title.

Line 68 – The aim to quantify the impact of "groundwater table drawdown" on hyporheic exchange processes. Replace "groundwater table drawdown" with "the degree of gaining/losing conditions".

Response: Please refer to the response to the first comment above.

Fig 3 – "Effect of diel river temperature fluctuations and daily groundwater table drawdowns on hyporheic fluxes…". Again daily groundwater table drawdown was not simulated, but daily fluctuations in gaining/losing conditions of the river.

Response: Please refer to the response to the first detailed comment above. Additionally, simulating groundwater table drawdown is not within the research scope. The research objective is to simulate hyporheic responses to daily

groundwater drawdown. Therefore, groundwater flux time series representing daily groundwater table drawdown were used directly.

Line 300 – "The timing of groundwater table drawdown also affects hyporheic exchange rates." Upward and downward fluxes representing gaining/losing conditions affects hyporheic exchange rates. There is a conceptual disconnect here.

> Response: Here we particularly refer to the timing of the lowest groundwater table. "Gaining and losing" can only represents the direction of the groundwater flow but not the temporal pattern of the groundwater fluctuation.

Line 310 – "Therefore, the timing of the aquifer pumping can potentially amplify or reduce the dispersal of pollutants in the aquifer". It would be better to state that the 'timing of river gaining/losing magnitude can potentially amplify or reduce…"

> Response: Here we specifically refer to the timing of the lowest groundwater fluxes in a hypothetical wastewater discharge event. "Timing of river gaining/losing magnitude" cannot accurately describe the timing of the lowest groundwater fluxes. Together with the response made above to the first detailed comment, we decided to keep the "time of aquifer pumping". Also, this paragraph is part of the results implication. Although aquifer pumping is not simulated but groundwater flux is directly used, the results have important implications on selecting pumping scheme with necessary considerations of river hydrologic conditions.

Lines 313 to 317 - Sure, but it also depends on the level of hydraulic connection between the river and the well (Barlow and Leake, 2012).

> Response: As explained above the hydraulic connection is beyond the research scope.

Line 380 – Scheduling pumping activities to protect thermal heterogeneity across multiple spatial scales again depends on the well-stream connection, if any.

> Response: Yes, it depends on the well-stream connection. However, the objective is not simulating groundwater fluctuations but rather the hyporheic response to various groundwater fluctuation scenarios. With the current model setting, the well-stream connection is irrelevant.

Line 392 – "hyporheic denitrification potential can be regulated by adjusting the timing of daily groundwater table drawdown." It would be better to state the "timing of hydraulic gradients towards the river or groundwater in/out flow to the river" and not "timing of daily groundwater table drawdown". The influence of groundwater table drawdown on the state of the river depends on its hydraulic connection and distance between the well and the river. It's possible to have a delayed response in the rivers condition or no response at all if there is little to no connection or if the well is very far from the river.

> Response: Please refer to the response to the first comment above.

Line 456 – "Groundwater table dynamics" replace with "Groundwater discharge/recharge to/from rivers substantially…"

> Response: Here the groundwater table dynamics not only include the direction of the groundwater flow (gaining or losing), but also the phase and amplitude of the groundwater table fluctuations. Using "groundwater discharge/recharge to/from rivers" is not accurate.

Line 471 – "…hyporheic denitrification potential is also changing following groundwater table drawdown." Replace "groundwater table drawdown" with "groundwater discharge/recharge". For the statement made be the authors about groundwater tables, one would require a 3D model that contains areas beyond the river banks. Instead, what has been simulated is a 2D section along the river with a prescribed flux boundary representing fluctuating gaining/losing conditions. There is a conceptual gap between the prescribed

boundary condition simulated in this study and groundwater table drawdowns that the authors describe. Groundwater table drawdowns could be happening at some distance away from the river with varying hydraulic properties between, which would dramatically influence the degree to which aquifer pumping would effect gaining/losing conditions at the river.

Response: As we explained in the response to comment on line 380, understanding groundwater responses to pumping activities is not our research objective. To avoid misunderstanding, we added the following text in Method section line 165:

*The objective of this study is not to understand groundwater responses to pumping activities. Even though the timing of groundwater table drawdown depends on multiple factors, i.e. hydrological connectivity between wells and aquifer, aquifer properties for plant water-use, and pumping capacity and electricity tariff for anthropocentric pumping activities, the two special cases, namely in-phase and out-of-phase groundwater conditions, can capture the representative dynamic hyporheic responses to different timing of daily groundwater withdrawal under corresponding river temperature conditions*

2. Model parameters

I am trying to gauge the realism of the model parameters but am having a hard time finding certain values. I have listed the specific line numbers for parameter values that were not stated or difficult to find. Clearly stating the parameter values would help future researchers to extend or repeat these numerical experiments.

Response: In the revised manuscript, we added the missing information. Please also find it below.

Line 91 – What were the values selected for permeability and porosity?

Response: This sentence is revised as below (line 91):

$\theta$ *is porosity 0.3[-], $\kappa$ is permeability [$L^2$] 1E-10 $m^2$.*

Line 100 – Do you have references to support this aspect ratio?

> Response: References are added as below (line 101)

> *In the present study, an aspect ratio (the ratio between amplitude and wavelength Δ/λ) of 0.1 and slope of 0.01 are used to describe the geomorphological setting as dunes (Dingman, 2009; Bridge, 2009).*

Line 108 – What is the hydrodynamic thermal dispersion tensor value chosen?

> Response: The detailed calculation steps of thermal dispersion tensor can be found in Wu et al. (2020). The reference was added (line 109):

> *$D_T$ is the hydrodynamic thermal dispersion tensor [$L^2 T^{-1}$] calculated following Wu et al. (2020).*

Line 288 – "mainly due to the change of hydraulic conductivity which is a function of diel temperature fluctuations." Again what was the permeability value chosen? How much does hydraulic conductivity vary in your simulations due to temperature? Line 341 states that there is a 220% change in K due to a 30 degree change in temperature, but I'm not sure if this is a realistic magnitude of daily temperature change or K change. Thomas (2014) shows temperature fluctuations over a year in Sauk River, Washington. There is a 18 degree C range over the year, but at the daily scale there is rarely more than a 10 degree fluctuation throughout the day. Perhaps the daily temperature range should be decreased by 3x, unless the authors have references that support a daily 30 degree temperature change in river water.

> Response: A permeability value was added in the revised manuscript and in the response to the comment on line 91 above. 30-degree change is the seasonal variation. As the reviewer wrote, the daily variation is much smaller. The calculation is based on 5-year river discharge datasets. It is not a result based on daily river temperature variations.

3. Other comments

Line 28 – Chow et al. (2019) conducted a sensitivity analysis on the effects of river bathymetry (i.e., geomorphological settings) on meander-scale hyporheic exchange.

Response: We have added the reference in the revised manuscript (line 29).

Line 30 – Chow et al. (2020) evaluated sediment heterogeneity and its effects on meander-scale hyporheic exchange.

Response: We have added the reference in the revised manuscript (line 32).

Line 40 – "Large groundwater upwelling and downwelling may compress 'or extend' hyporheic…"

Response: According to our definition of hyporheic zone following Triska et al. (1989) and Gooseff (2010), and the flow reverse technique that is commonly used in simulating losing conditions (line 139), large groundwater upwelling and downwelling will always compress hyporheic zons. For strong gaining conditions, less surface water can penetrate the subsurface; for strong losing conditions, more surface water will flow downwards without returning to the surface. Therefore, less hyporheic exchange occurs under larger gaining or losing conditions.

Fig 1b and c – The hyporheic exchange should be compressed in Fig 1b and extended in Fig 1c. Instead it looks the same between Fig. 1b and c.

Response: The size of the hyporheic zone depends on the hydraulic gradient at the sediment-water interface and not only on the direction of the groundwater flow. Hyporheic zones under gaining conditions are not necessarily larger than that under losing conditions. Figure 1 is only a conceptual description and not supposed to be used to compare the hyporheic zone extension. Determining accurate size of hyporheic zone extension requires numerical modeling.

Lines 324 to 330 – I find this sentence confusing. So what plays a dominant role in the winter? Pumping? Please clarify.

Response: The following sentence was added to better describe the effect of river temperature (line 331):

*In other words, higher river temperature has larger impacts on the temporal variations of hyporheic exchange.*

Line 380 – Remove s from 'cares'

Response: Done as suggested (line 381).

Line 387-389 – Wouldn't losing conditions have longer residence times since the flow paths would be extended and stretched?

Response: Under gaining condition, there is mixing between surface water and groundwater which has significantly longer residence time than surface water. Therefore, the overall residence time under gaining condition is generally longer than under losing condition where there is no groundwater mixing.

Line 467 to 468 "The timing of aquifer pumping should be adjusted to avoid…". Can you be more specific here. I.e., "The pumping should decrease or stop during flood events in order to ensure minimal contaminant uptake".

Response: This sentence was reworded as below (line 467):

*Pumping activities should be avoided during flood events in order to ensure*

*minimal contaminant uptake.*

Lastly, a general comment about the two scenarios of in-phase and out-of-phase compared throughout the manuscript. It would be nice to get from the authors their ideas on how likely these two scenarios are and what kinds of assumptions must be met in order for them

to be realistic. I can imagine that in-phase is a more likely scenario because ET tends to increase as temperatures increase during the day. Also, in cases where there is hydraulic connection and a short distance between the well and river, I would expect that groundwater usage would increase following daytime activity. Out-of-phase may be less likely, but this could depend on the well-stream connection.

> Response: Groundwater uptake by vegetations is higher with higher air/river temperature during the day. Additionally, because of the large storage capacity of the modern reservoir, pumping activities can be scheduled independent of the human need. Therefore, pumping can happen at any time during the day including the timing of the two special scenarios explored in this study.

**References**

Barlow, P.M. and Leake, S.A., 2012. Streamflow depletion by wells: understanding and managing the effects of groundwater pumping on streamflow (p. 84). Reston, VA: US Geological Survey.

Bridge, J. S.: Rivers and floodplains: forms, processes, and sedimentary record, John Wiley & Sons, 2009.

Chow, R., Bennett, J., Dugge, J., Wöhling, T. and Nowak, W., 2020. Evaluating subsurface parameterization to simulate hyporheic exchange: The Steinlach River Test Site. Groundwater, 58(1), pp.93-109.

Chow, R., Wu, H., Bennett, J.P., Dugge, J., Wöhling, T. and Nowak, W., 2019. Sensitivity of simulated hyporheic exchange to river bathymetry: The Steinlach River test site. Groundwater, 57(3), pp.378- 391.

Dingman, S. L.: Fluvial Hydraulics, Oxford University Press, USA, Oxford ; New York, 2009.

Thomas, L., 2014. Stream Temperature Variability: Why It Matters to Salmon. Science Findings, United States Department of Agriculture Forest Service. https://www.fs.fed.us/pnw/sciencef/scifi163.pdf

Wu, L., Gomez-Velez, J. D., Krause, S., Singh, T., Wörman, A., and Lewandowski, J.: Impact of flow alteration and temperature variability on hyporheic exchange, Water Resources Research, 2020.